# Classification of Shoulder X-ray Images with Deep Learning Ensemble Models

Fatih Uysal [1],*, Fırat Hardalaç [1], Ozan Peker [1], Tolga Tolunay [2] and Nil Tokgöz [3]

[1] Department of Electrical and Electronics Engineering, Faculty of Engineering, Gazi University, TR 06570 Ankara, Turkey; firat@gazi.edu.tr (F.H.); ozan.peker@gazi.edu.tr (O.P.)

[2] Department of Orthopaedics and Traumatology, Faculty of Medicine, Gazi University, TR 06570 Ankara, Turkey; tolgatolunay@gazi.edu.tr

[3] Department of Radiology, Faculty of Medicine, Gazi University, TR 06570 Ankara, Turkey; nil.tokgoz@gazi.edu.tr

* Correspondence: uysal@gazi.edu.tr; Tel.: +90-534-022-6128

**Abstract:** Fractures occur in the shoulder area, which has a wider range of motion than other joints in the body, for various reasons. To diagnose these fractures, data gathered from X-radiation (X-ray), magnetic resonance imaging (MRI), or computed tomography (CT) are used. This study aims to help physicians by classifying shoulder images taken from X-ray devices as fracture/non-fracture with artificial intelligence. For this purpose, the performances of 26 deep learning-based pre-trained models in the detection of shoulder fractures were evaluated on the musculoskeletal radiographs (MURA) dataset, and two ensemble learning models (EL1 and EL2) were developed. The pre-trained models used are ResNet, ResNeXt, DenseNet, VGG, Inception, MobileNet, and their spinal fully connected (Spinal FC) versions. In the EL1 and EL2 models developed using pre-trained models with the best performance, test accuracy was 0.8455, 0.8472, Cohen's kappa was 0.6907, 0.6942 and the area that was related with fracture class under the receiver operating characteristic (ROC) curve (AUC) was 0.8862, 0.8695. As a result of 28 different classifications in total, the highest test accuracy and Cohen's kappa values were obtained in the EL2 model, and the highest AUC value was obtained in the EL1 model.

**Keywords:** biomedical image classification; bone fractures; deep learning; ensemble learning; shoulder; transfer learning; X-ray

## 1. Introduction

Having a wider and more varied range of movement than the other joints in the body, the shoulder has a flexible structure. The fractures in the shoulder may result from incidents such as dislocation of the shoulder and engaging in contact sports and motor vehicle accidents. The shoulder bone mainly consists of three different bones: the upper arm bone, named the 'humerus', the shoulder blade, named the 'scapula', and the collarbone, named the 'clavicle'. Figure 1 shows the anatomic structure of the shoulder bone. The image in this figure was taken from the MURA dataset used in this study, and the markings thereon were placed by the physicians at Gazi University. As shown in Figure 1, the upper end of the humerus has a ball-like shape that connects with the scapula, called the glenoid. The types of shoulder fractures vary depending on age. While most fractures in children occur in the clavicle bone, the most common fracture in adults occur on the top part of the humerus, i.e., the proximal humerus. The types of shoulder bone fractures are divided into three categories in general: clavicle fractures, which are the most common shoulder fracture, frequently the result of a fall, scapula fractures, which rarely occur, and resulting fractures, which occur as cracks in the upper part of the arm in individuals over 65 years of age. The images from X-ray devices are primarily used for imaging of the shoulder bone for diagnosis and treatment of such fractures, while MRI or CT devices may also be used when required [1].

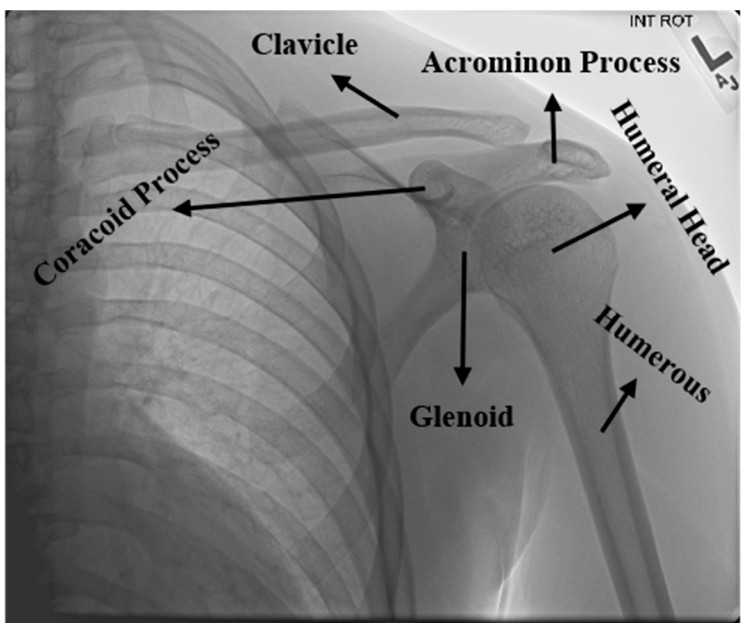

**Figure 1.** The anatomy of the shoulder bone.

Deep learning classification procedures of shoulder bone X-ray images were carried out in the study. The main contributions of this study are as follows:

- The most suitable model for the classification of shoulder bone X-ray images as a fracture or non-fracture is determined.
- An approach that can be used in similar studies is developed via new ensemble learning models.
- The study can assist physicians who are not experts in the field in the classification stage, especially in cases of shoulder fractures, which are frequently encountered in the emergency departments of hospitals.
- The method suggested in the study contributes to the literature with two different ensemble approaches.
- With the first model proposed in the study, a performance study was conducted with transfer learning for the MURA dataset, which is widely used in X-ray studies. Thus, three models that give the best classification results have been combined into a single model, and the classification performance has been increased. The developed model can be used to classify many medical X-ray images.
- With the second model, it is determined which model finds which class best by looking at the success of finding the classes on the models in the dataset. It is ensured that the model decides the prediction of that class. Thus, regardless of the dataset studied, a similar decision system can be designed with models that find the classes in a given dataset, and a higher performance can be achieved compared to a single model.
- The proposed ensemble model approaches can be applied and generalized by performing similar preprocessing steps in other X-ray biomedical datasets. In addition, the proposed method can be easily used in studies with transfer learning.

## 2. Related Works

In this study, the classification of X-ray images of shoulder fractures in the MURA dataset was carried out into the categories of 'fracture' or "non-fracture' using built CNN-based deep learning models, new models thereof adapted to SpinalNet, and models developed based on ensemble learning. Therefore, two main topics have been taken into account while examining the literature regarding the study. Upon examination of the studies conducted thus far, these topics are as follows:

- what the studies conducted using the MURA dataset are, and why this dataset is preferred in this study;
- what kinds of studies have been conducted on the classification of shoulder bone images, and what kinds of innovations may be set forth based on the efficiency of deep learning models used in the classification procedures carried out in this study.

### 2.1. Studies Conducted Using the MURA Dataset

The MURA dataset was first introduced to the literature in a paper published in the OpenReview platform, announced in the conference on "Medical Imaging with Deep Learning" held in Amsterdam in 2018. Following this publication, this dataset was made publicly available for academic studies in a competition called "Bone X-Ray Deep Learning Competition" by the Machine Learning group of the Stanford University. Being one of the largest public radiographic image datasets, MURA contains a total of 40,561 X-ray images in png format for the following parts of the body, labeled as either normal or abnormal (fracture): elbow, finger, forearm, hand, humerus, shoulder, and wrist. In this classification study conducted by Rajpurkar et al. using DenseNet-169 on this dataset, the AUC score representing the area under the overall Receiver Operator Characteristics (ROC) curve was 0.929, and the overall Cohen's kappa score was 0.705 [2]. Following this first study in which this dataset was introduced to the literature, there have been various studies that have been conducted and published using all or a part of the dataset. These studies are as follows: an average precision (AP) value of 62.04% was achieved in the fracture detection procedure using the proposed deep CNN model after the fractures were marked by the physicians on the arm X-ray images in the MURA dataset by Guan et al. [3]. The following classification accuracies were achieved by Galal et al. using a very small part of the elbow X-ray images in the MURA dataset (56 images for the train dataset, 24 images for the test): 97% with the support vector machine (SVM), 91.6% with random forest (RF), and 91.6% with naive Bayes [4]. The classification by Liang and Gu with the multi-scale CNN and graph convolution network (GCN) they proposed using the entire MURA dataset achieved an overall Cohen's kappa score of 0.836. [5]. The overall Cohen's kappa score achieved by Saif et al. in classification performed using the whole MURA dataset with the proposed capsule network architecture was 0.80115 [6]. In the classification carried out by Cheng et al. using the dataset containing hip bone images and the whole MURA dataset, the highest accuracy achieved was 86.53% for humerus images with the proposed adversarial policy gradient augmentation (APGA) [7]. The highest accuracy achieved in the classification performed by Pelka et al. using the whole MURA dataset was 79.85% with the InceptionV3 model [8]. The AUC score was 0.88 in the classification carried out by Varma et al. on a Lower Extremity Radiographs Dataset (LERA) containing foot, knee, ankle, and hip data for an ImageNet and DenseNet161 model pre-trained with MURA [9]. In the classification performed by Harini et al. on images of the finger, wrist, and shoulder in the MURA dataset using five different CNN-based built deep learning methods, the highest accuracy was 56.30% on the wrist data with DenseNet169 [10]. The overall accuracy achieved for MURA with the proposed iterative fusion CNN (IFCNN) method is 73.4% in classification carried out by Fang et al. using the optical coherence tomography (OCT) dataset and the entire MURA dataset [11]. The overall Cohen's kappa score achieved was 0.717 with the proposed deep CNN-based ensemble model in the classification carried out by Mondol et al. on elbow, finger, humerus, and wrist data in the MURA dataset [12]. In the type classification performed by Pradhan et al. using the whole MURA dataset, 91.37% accuracy was achieved with the proposed deep CNN model [13]. In the classification carried out by Shao and Wang using the entire MURA dataset, a two-stage method was developed, and the highest accuracy achieved was in humerus images with 88.5% for the SENet154 model, while the highest accuracy for the DenseNet201 model was achieved again on humerus images with 90.94% [14].

Although there are many different publicly available datasets on bone fractures in addition to the MURA dataset in the literature, the main reason for using only this dataset

in this study is that MURA is one of the largest datasets for both normal (negative, non-fracture) and abnormal (positive, fracture) groups compared to the other public datasets. It is observed upon examination of the studies conducted via this dataset that classification and/or fracture detection procedures are carried out using the dataset in whole or in part. Only the X-ray images for shoulder bone in the MURA dataset were used in this study. The reason why only this type of dataset was used for classification despite the availability of seven different types of datasets and the reason why such a dataset contains shoulder data is that the shoulder dataset is the most balanced type in terms of distribution of the amount of data provided for both training and validation. Another reason for classification for a single type only is to be able to develop the most optimum model for fracture and non-fracture images on this type, using and developing as many different deep learning models as possible.

### 2.2. Classification Studies Carried Out on the Shoulder Bone

While some of the studies mentioned under the previous title with the MURA dataset include classification of shoulder bone images, there are also studies in the literature on this type of classification exclusively. In classification carried out by Chung et al. with the ResNet152 model on a total of 1891 patients, including 1376 proximal humeral fracture cases of four different types, a 96% accuracy was achieved [15]. In the CNN model proposed by Sezers using a total of 219 shoulder MR images, a classification was made in three different groups as normal, edematous, and Hill–Sachs lesions with a 98.43% accuracy [16]. The highest accuracy achieved in the classification carried out by Urban et al. on 597 X-ray images of shoulders with implants was 80.4% in the NASNet model pre-trained with ImageNet [17]. In the classification performed by Sezer et al. using a total of 219 shoulder MR images grouped as 91 edematous, 49 Hill–Sachs lesions, and 79 normal, an 88% success rate was achieved with a kernel-based SVM, and 94% success was achieved with extreme learning machines [18]. A 94.74% accuracy was achieved with the proposed CapsNet model in the classification performed by Sezers on a total of 1006 shoulder MR images, grouped as 316 normal, 311 degenerated, and 379 torn [19]. Some other important studies on medical data classification and machine/deep learning approach in the literature are as follows: An accuracy of 93.5% was achieved in cardiac arrhythmia classification procedures with long short-term memory (LSTM) by Khan and Kim [20]. ResNet18 and GoogLeNet models were used by Storey et al. to detect wrist bone abnormality [21]. As a result of the classification made by Yin et al. for kidney disease with the multi-instance deep learning method, an accuracy of 88.6% was obtained [22]. As a result of the detection of musculoskeletal abnormalities performed by Dias, the highest accuracy was obtained as 81.98% with the SqueezeNet model [23]. An accuracy of 98.20% was achieved by Khan and Kim using spark machine learning and conv-autoencoder to detect and classify unpredictable malicious attacks [24]. Transcriptional co-activators have been identified by Kegelman et al. as regulators of bone fracture healing [25]. The bone fracture process performed by Sharma et al. with machine learning and digital geometry achieved 92% classification accuracy [26].

It is observed upon examination of the recent classification studies on shoulder bone in the literature that the normal and abnormal (fracture, edematous, Hill–Sachs lesion, degenerated, or torn) images obtained from CT, MRI, or X-ray devices are classified by not only the traditional machine learning methods such as the SVM but also by deep learning-based methods such as ResNet, NasNet, and CapsNet. In this study, the classification was performed on normal and abnormal shoulder bone X-ray (fracture) images in the MURA dataset. In the performance thereof, as a practice different from the literature, built CNN-based deep learning models (ResNet, ResNeXt, DenseNet, VGG, InceptionV3, and MobileNetV2) with a varying structure and number of layers were used by transfer learning with pre-trained ImageNet by also adding SpinalNet to the classification layers. Based on the results of classification achieved by the procedures performed herein, another classification was performed with ensemble learning. The main reason for applying

transfer learning on built CNN models, adding SpinalNet, and applying ensemble learning is to contribute to the efficiency of deep learning models proposed in the classification of shoulder bone X-ray images.

Section 3 discusses the CNN-based built deep learning models, such as ResNet, DenseNet, VGG, and InceptionV3, used in classification as well as their adaptations with Spinal FC and the proposed ensemble learning models. Section 4 explains the open source dataset, including the shoulder bone X-ray images, data augmentation, and data preprocessing procedures applied thereon, a table containing the accuracy, recall, F1-score, and Cohen's kappa scores achieved by classification models, and the results of classification, including a confusion matrix, ROC curves, and AUC scores. The last section interprets the results achieved by classification models and discusses the contribution of this study to the literature and improvements that can be applied in future work.

## 3. Methods

A number of different deep-learning-based methods have been used and developed to classify X-ray images of the shoulder bone in png format and with three channels as normal and abnormal in this study. Firstly, classification was performed with ResNet (34,50,101,152), ResNeXt (50,101), DenseNet (169,201), VGG (13,16,19), InceptionV3, and MobileNetV2, which are CNN-based deep learning models with different structures and layers that are publicly available. Subsequently, new classification networks were established by replacement of the classification layer of each model used herein with the SpinalNet FC layer. Based on the results obtained herein, new ensemble learning models specific to this study were developed in order to further increase the classification accuracy. Built deep learning models, newly developed models with SpinalNet, and details of the ensemble learning models used for classification are described in the following subsections.

### 3.1. Classification Models Based on CNNs for Shoulder Bone X-ray Images

The CNN-based deep learning models currently available were used to classify X-ray images of the shoulder bone. In training the models, training was firstly carried out with data in the training section of the dataset without pre-training, i.e., with a random weight as the initial weight. However, the required level could not be reached in training the network or at the end of classification by this application. Therefore, the transfer learning method was applied.

In this transfer learning method, two versions of each available deep learning model pre-trained with ImageNet data, i.e., the existing weights in the pre-trained models, were used. ImageNet is a dataset and benchmark that contains millions of images and hundreds of object categories, on which the procedures of image classification, single-object localization, and object detection are mainly performed [27]. The shoulder bone X-ray images dataset used in this study was used for input of the deep learning models, which are specified in Figure 2 and were trained with ImageNet data. The structure comprised of 1000 classes in the last layer of these models was made to have two classes in order to classify bone images specific to our study as normal and abnormal. Following the abovementioned procedures, classification was carried out by training the network with the shoulder dataset. Moreover, the last layer of 13 built deep learning models were fine-tuned with SpinalNet/Spinal FC; with the newly acquired models, shoulder bone X-ray images were trained and classified.

The currently available ResNet, ResNeXt, DenseNet, VGG, InceptionV3, and MobileNetV2 deep learning models, which can be used in classification, along with versions with different numbers of layers were some of the models used in the classification of shoulder bone X-ray images in this study. The required method-based details regarding these models and SpinalNet are described in the following subsections.

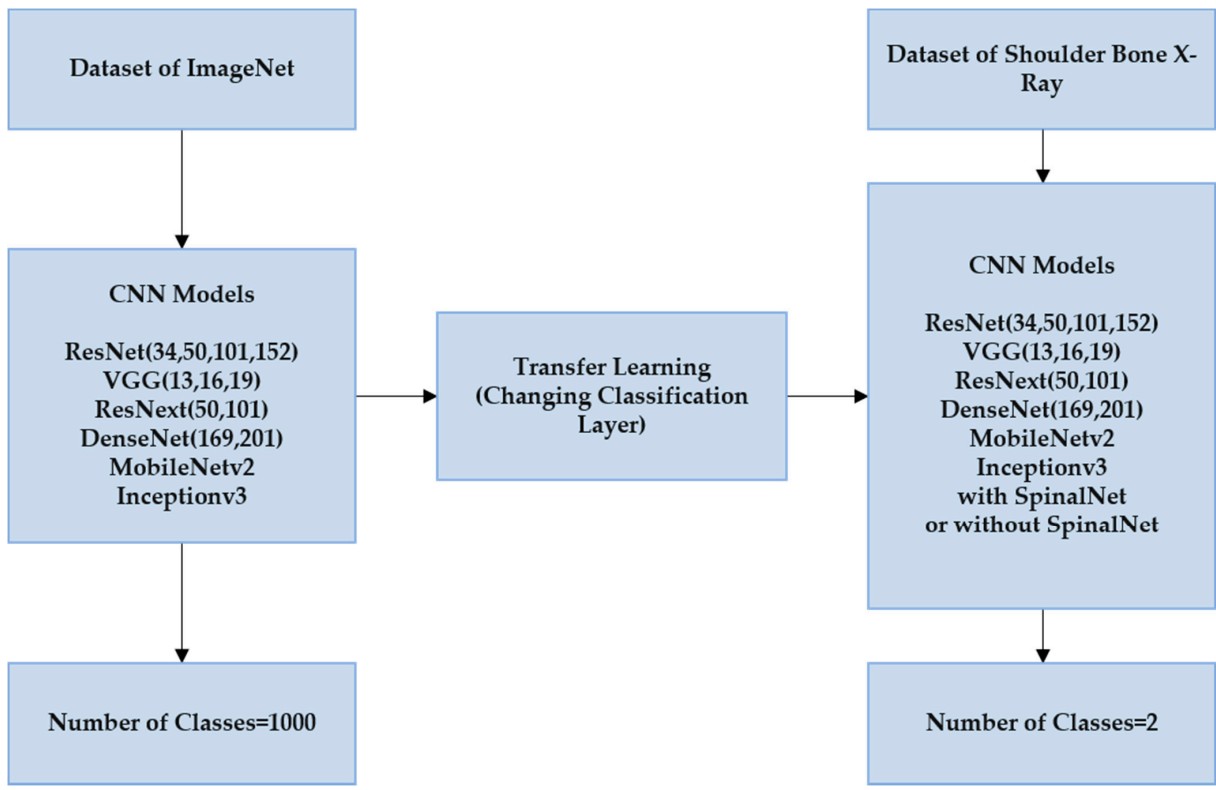

**Figure 2.** Standard FC/Spinal FC CNN-based deep learning methods subject to transfer learning.

### 3.1.1. ResNet

ResNet is a CNN-based deep learning model including more than one building blocks of residual learning depending on the number of layers [28]. In part (a) of the Figure 3, there are k number of building blocks with $3 \times 3$ conv for ResNet34, and, in part (b), there are m number of building blocks with $1 \times 1$ and $3 \times 3$ conv and n number of building blocks with $1 \times 1$ conv for ResNet50, 101, and 152, respectively.

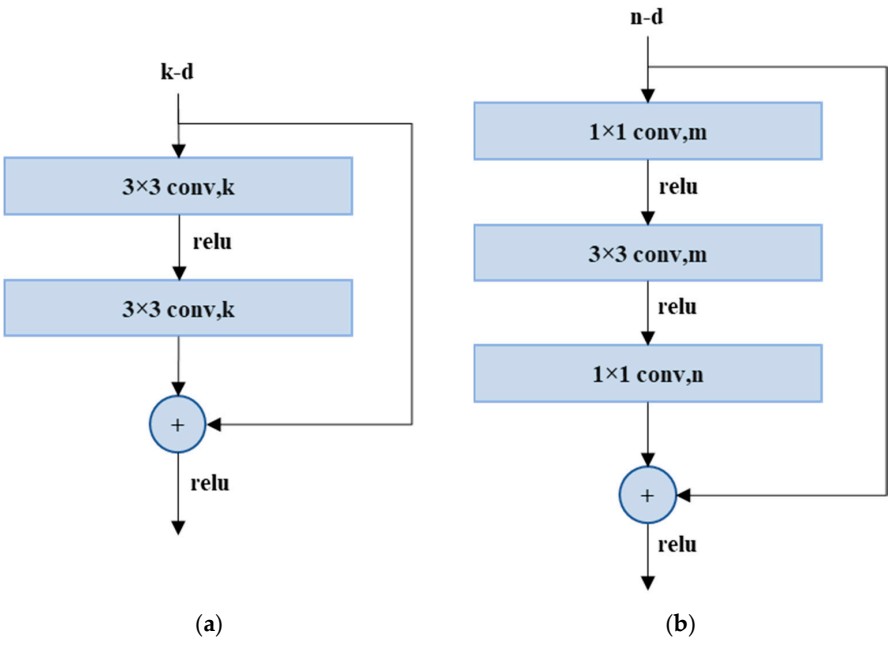

**Figure 3.** ResNet building block. (**a**) BB-I, k; (**b**) BB-II, m, n [28].

In this study, four ResNet models with 34, 50, 101, and 152 layers, respectively, were used in classification. The models were adapted, and the structure is shown in Figure 4.

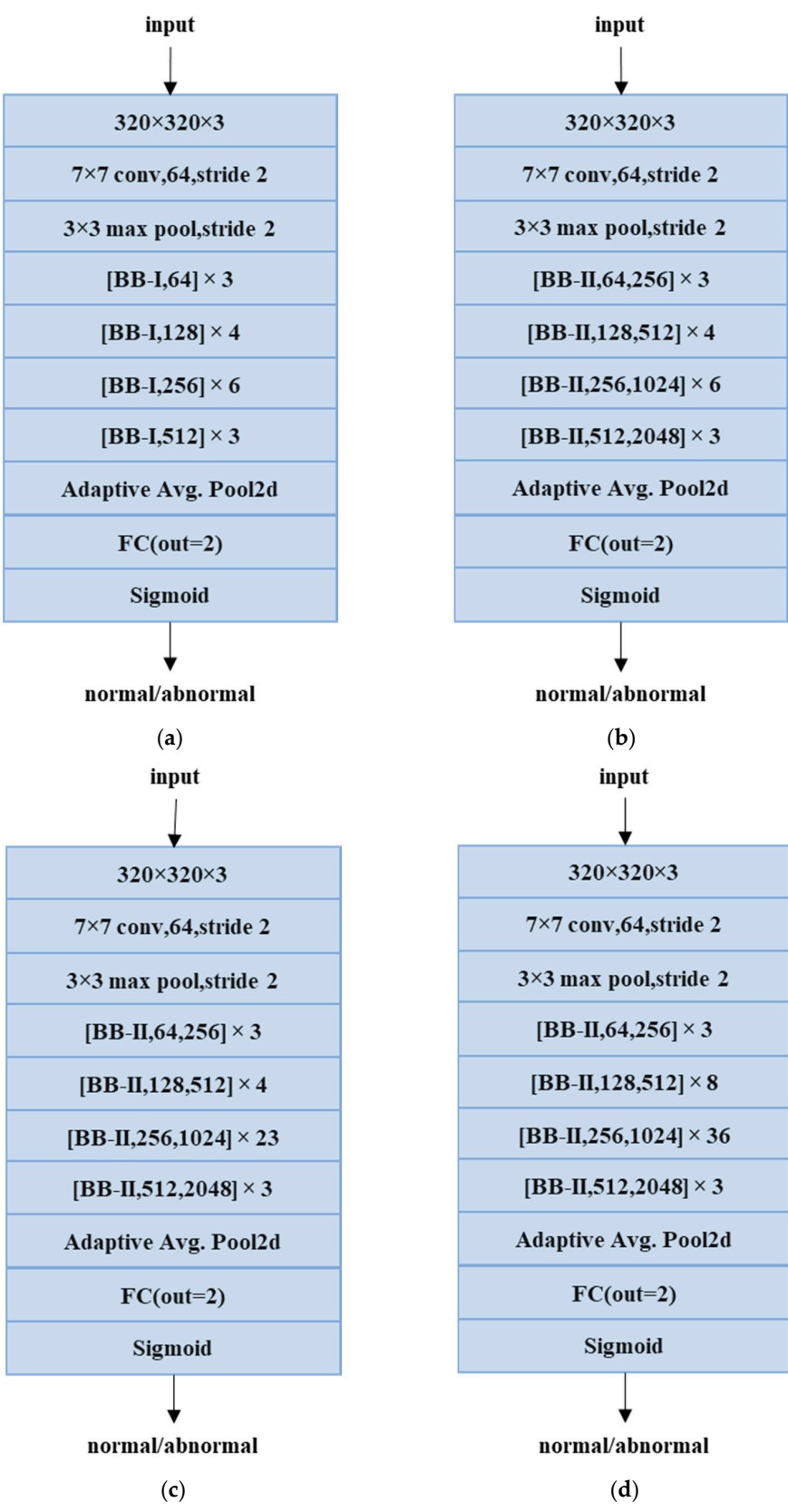

**Figure 4.** The structure of the adapted ResNet-based models. (**a**) ResNet34; (**b**) ResNet50; (**c**) ResNet101; (**d**) ResNet152.3.1.2. ResNeXt [28].

ResNeXt is a deep learning model for deep neural networks that reduces the number of parameters in ResNet. With this model, cardinality, which is an additional dimension for the width and depth of ResNet, defining the size of the set of transformations, is used [29]. The structure of the ResNext block is shown in Figure 5. In this structure, there is a block structure with a cardinality value of 32, with r number of $1 \times 1$ and $3 \times 3$ conv and s number of $1 \times 1$ conv, respectively.

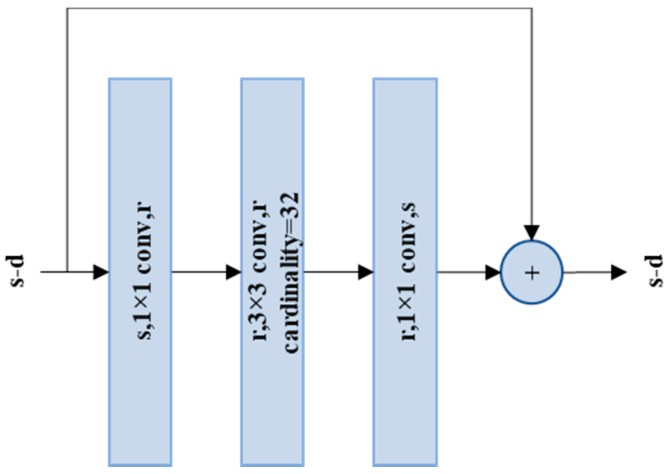

**Figure 5.** ResNeXt building block (BB-III, r, s, C = 32) [29].

Two ResNeXt models with 50 and 101 layers, respectively, were used in classification in this study. The models were adapted, and their structure is shown in Figure 6.

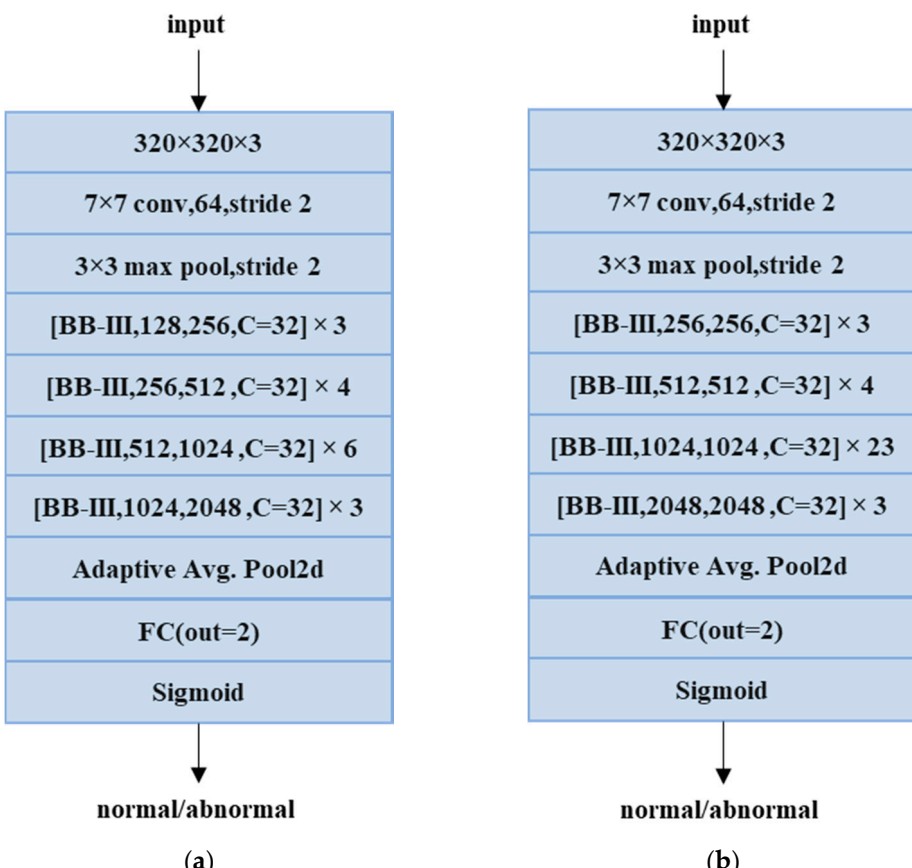

**Figure 6.** The structure of the adapted ResNeXt-based models. (**a**) ResNeXt50 (32 $\times$ 4d); (**b**) ResNeXt101 (32 $\times$ 8d) [29].

### 3.1.2. DenseNet

In a dense convolutional neural network, known as DenseNet, each layer affects every other layer in a feed-forward fashion [30]. A DenseNet block consists of five layers. The first four are dense layers, and the last is a transition layer. If the growth rate value is (k) for each layer, it is 4 for this dense block. In the transition layer, there is a $2 \times 2$ average pool with a $1 \times 1$ conv and a stride of 2. In the dense layer, there are $1 \times 1$ and $3 \times 3$ convs with a stride of 1.

In this study, two DenseNet models with 169 and 201 layers, respectively, and a growth rate (k) of 32 were used in classification. These models were adapted, and their structure is shown in Figure 7.

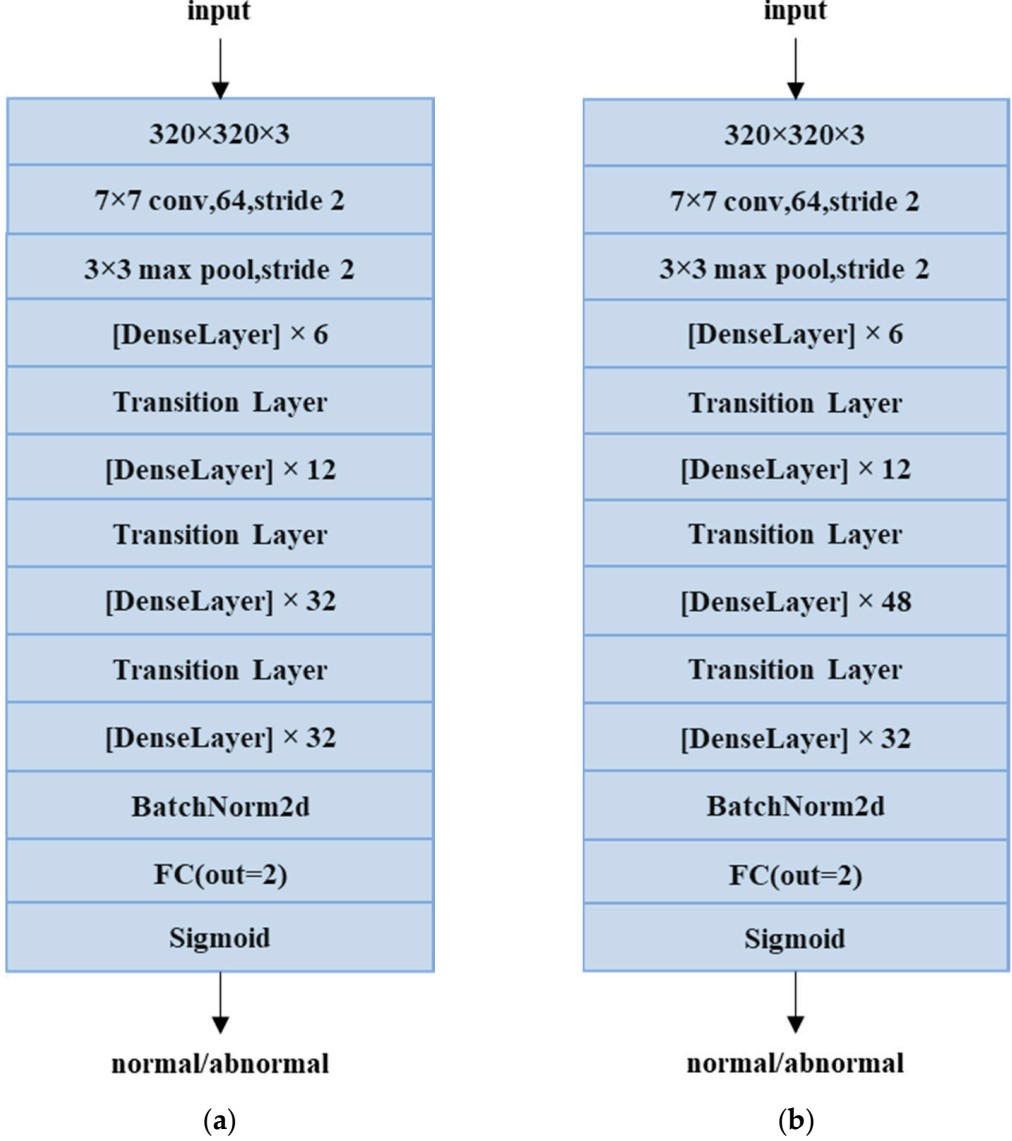

**Figure 7.** The structure of the adapted DenseNet-based models. (**a**) DenseNet169; (**b**) DenseNet201 [30].

### 3.1.3. VGG

VGG is a CNN-based deep learning model with very small ($3 \times 3$) convolution filters [31]. In this study, three VGG models with 13, 16, and 19 layers, respectively, were used for the classification procedure. The structure of conv blocks was comprised of a combination of 2, 3, or 4 consecutive $3 \times 3$ conv (with k, m or n filters) layers in Figure 8, and these VGG-based models, which were used and adapted from a combination thereof, are shown in Figure 9.

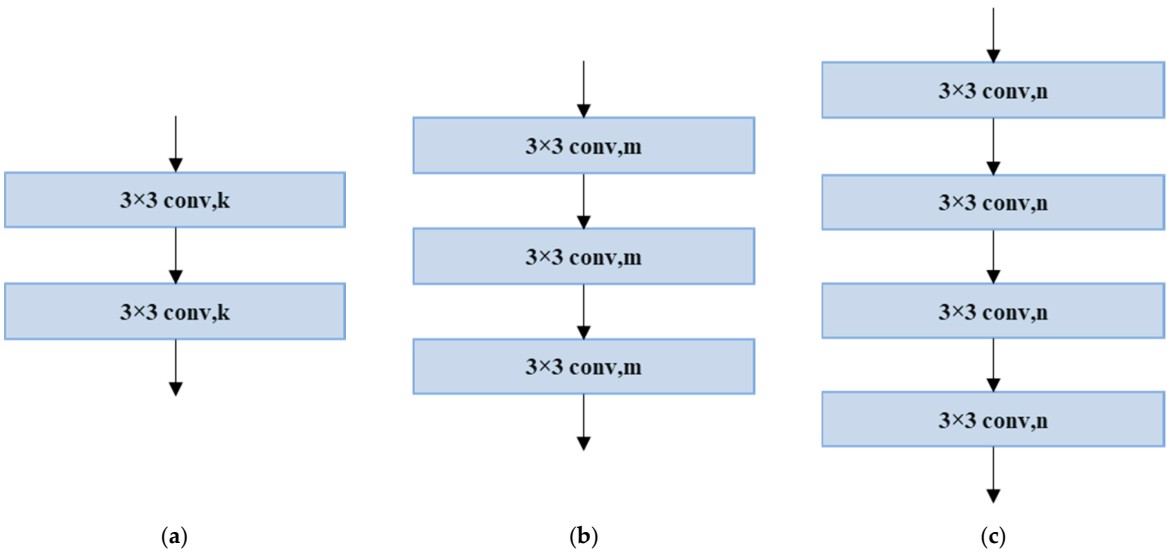

**Figure 8.** The conv blocks of 2, 3, and 4 3 × 3 conv layers (CB-I, II, and III, respectively). (**a**) CB-I, k; (**b**) CB-II, m; (**c**) CB-III, n [31].

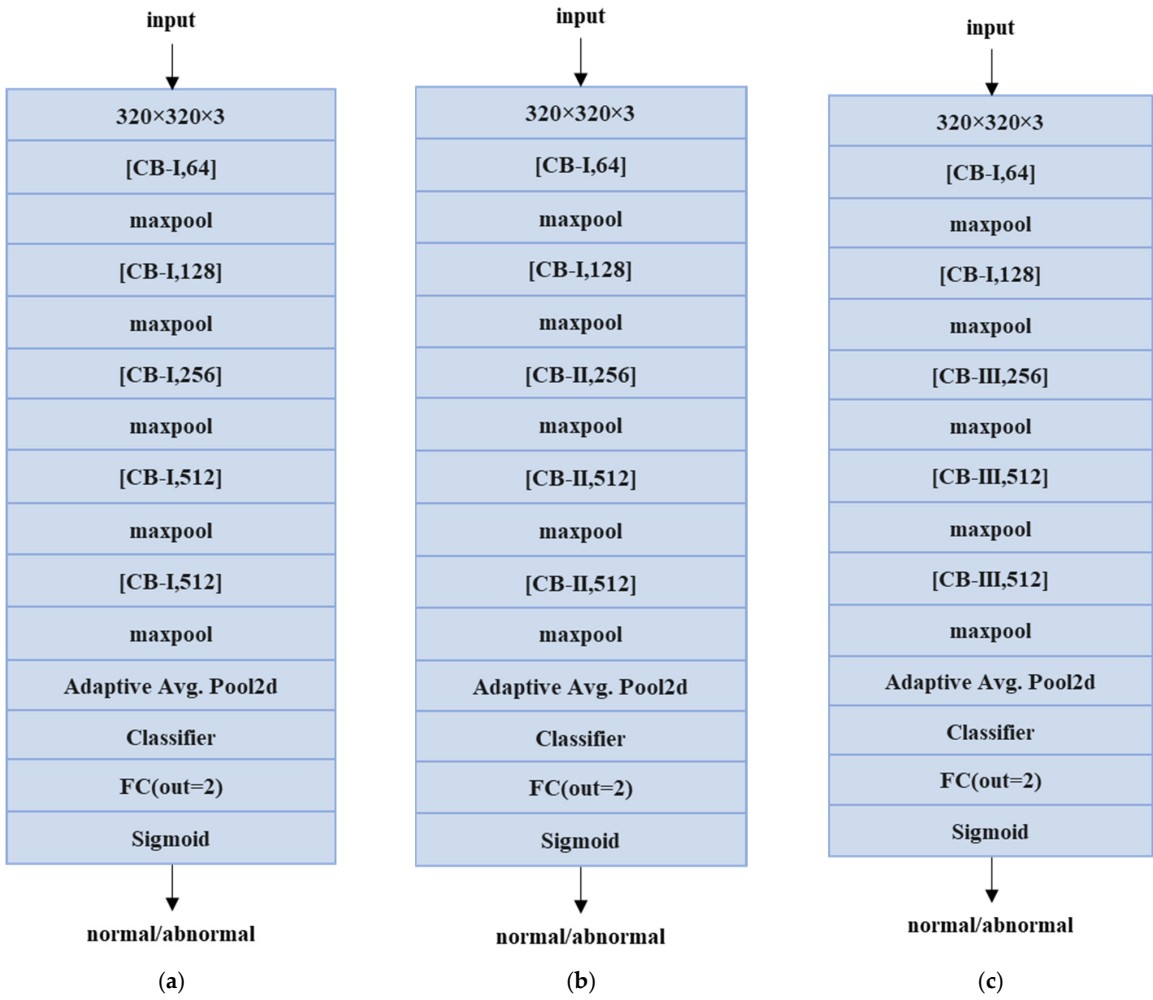

**Figure 9.** The structure of the adapted VGG-based models. (**a**) VGG13; (**b**) VGG16; (**c**) VGG19 [31].

### 3.1.4. InceptionV3

InceptionV3 is a CNN-based deep learning model that includes convolution factorization, which is called the inception module [32]. The structure is shown in Figure 10.

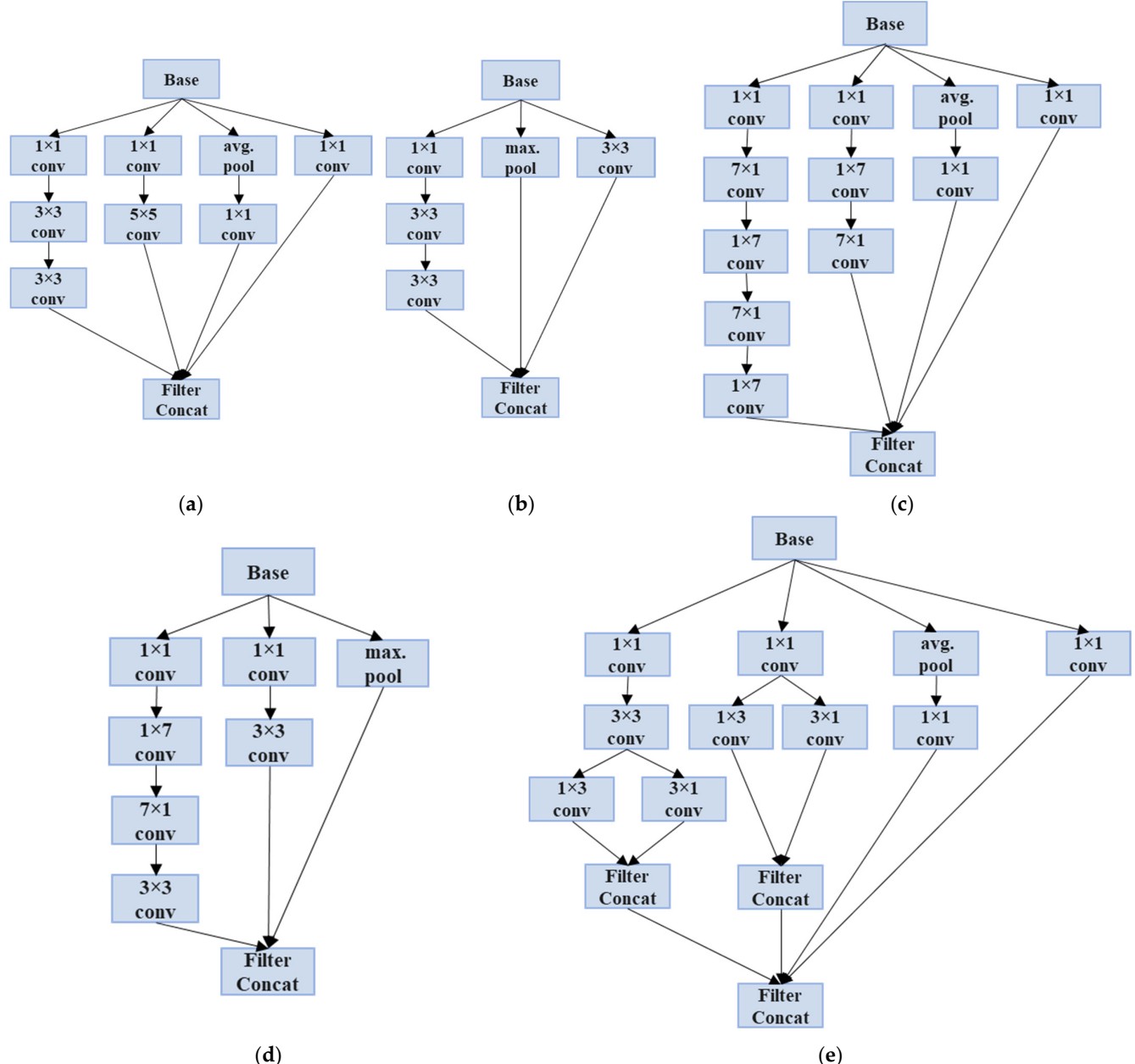

**Figure 10.** The inception blocks containing convolution factorization (IB-A, B, C, D, and E). (**a**) IB-A; (**b**) IB-B; (**c**) IB-C; (**d**) IB-D; (**e**) IB-E [32].

In this study, the InceptionV3 model was used in classification. This model was adapted, and its structure is shown in Figure 11.

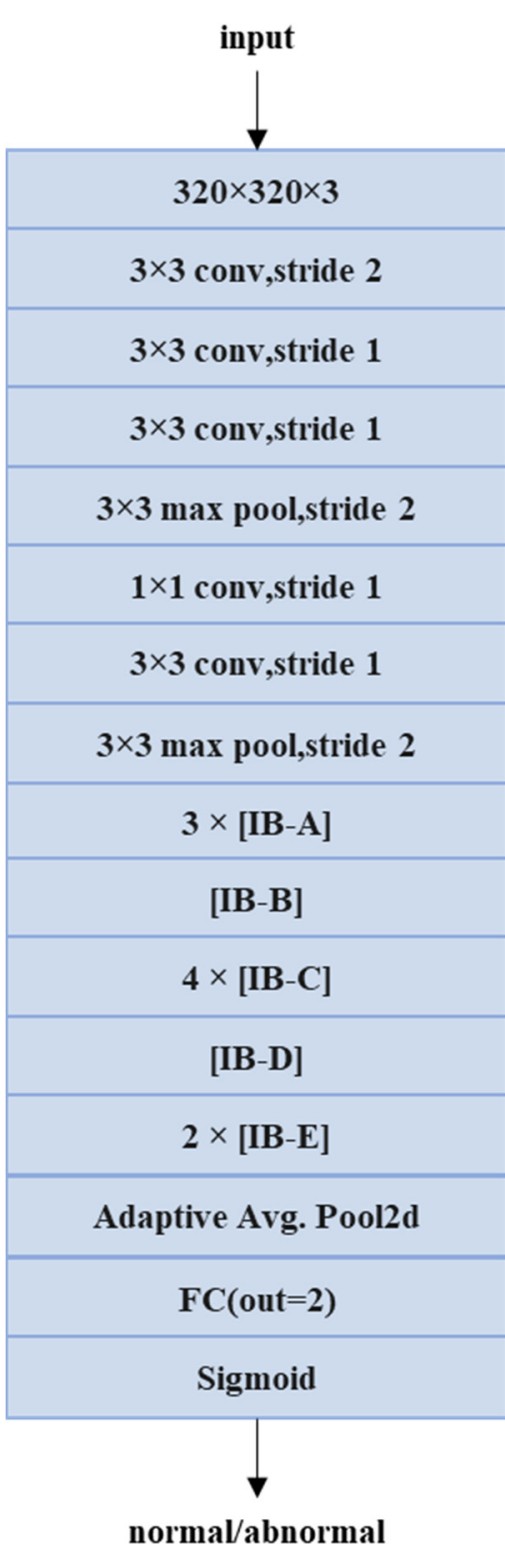

**Figure 11.** The structure of the adapted InceptionV3-based model [32].

### 3.1.5. MobileNetV2

MobileNetV2 is a deep learning model that includes residual bottleneck layers [33]. The structure of the convolution block of this model is shown in Figure 12.

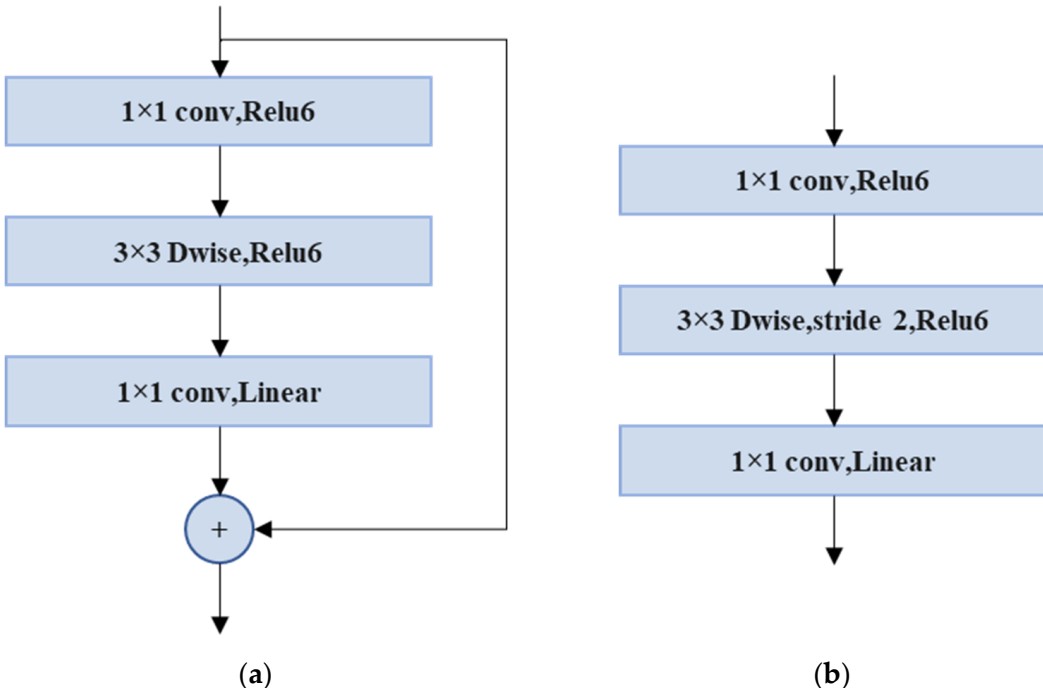

**Figure 12.** MobileNet bottleneck blocks where stride = 1 and stride = 2, respectively (MB-I and -II). (**a**) MB-I; (**b**) MB-II [33].

The MobileNetV2 model was used in classification in this study. The structure of this MobileNetV2-based model, which contains a total of 17 MobileNet bottleneck blocks, comprised of 13 bottlenecks with a stride value of 1, and 4 bottlenecks with a stride value of 2, is shown in Figure 13.

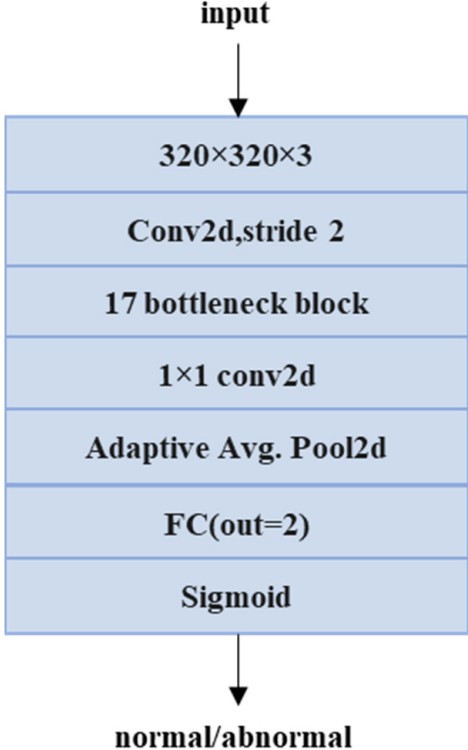

**Figure 13.** The structure of the adapted MobileNetV2-based model [33].

### 3.2. SpinalNet

In SpinalNet, the structure of the hidden layers is different from a normal neural network model. In a neural network, the hidden layers receive inputs in the first layer and then transfer the intermediate outputs to the next layer. However, the hidden layer in SpinalNet enables the previous layer to receive a certain part of its inputs and outputs. Therefore, the number of incoming weights in the hidden layer is lower than in normal neural networks [34].

The classification layer of the 13 above-mentioned CNN models based on ResNet, ResNeXt, DenseNet, VGG, InceptionV3, and MobileNetV2, used in classification, was adapted with SpinalNet/Spinal FC, and classification was also carried out with Spinal FC versions of these models. This allowed observation of the effect of SpinalNet on the classification of shoulder bone X-ray images. The details regarding this effect are described in Section 4. Moreover, the layer widths in Spinal FC of each classification model are as Table 1.

**Table 1.** Layer with values of Spinal FCs used in classification models.

| Models | Spinal FC Layer Width | Models | Spinal FC Layer Width |
|---|---|---|---|
| ResNet34 | 256 | ResNeXt50 | 20 |
| ResNet50 | 128 | ResNeXt101 | 128 |
| ResNet101,152 | 1024 | DenseNet169, 201 | 240 |
| VGG13 | 256 | MobileNetV2 | 320 |
| VGG16,19 | 512 | InceptionV3 | 20 |

### 3.3. Proposed Classification Models Based on EL for Shoulder Bone X-ray Images

Two ensemble models were established to further increase the accuracy of classification by examining the classification results of shoulder bone X-ray images performed with the CNN-based models and their SpinalNet versions used in the study, mentioned in the previous subsections. While choosing the sub-models required for ensemble learning, the accuracy rates in the classification results of each model, the confusion matrix scores, and the (normal/abnormal) AUC scores for each class were taken into consideration. The details of these two ensemble-based classification models are explained in the following subsections.

3.3.1. EL1 (ResNext50 with Spinal FC, DenseNet169 with Standard FC, and DenseNet201 with Spinal FC)

The EL1 (Ensemble learning-1) model was developed using a combination of ResNext50 with Spinal FC, DenseNet169 with Standard FC, and DenseNet201 with Spinal FC. A schematic diagram of the developed model is shown in Figure 14.

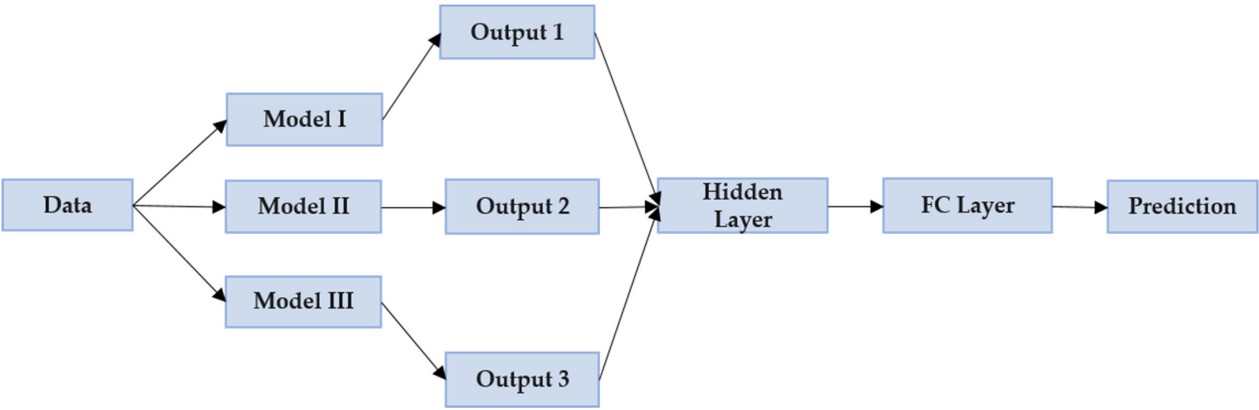

**Figure 14.** The proposed EL1 classification model.

It is observed upon examination of the figure above of the EL1 model proposed for classification of shoulder bone X-ray images as normal/abnormal (fracture) that three different CNN-based models are used. The "data" specified in this figure represent the entire dataset of shoulder bone X-ray images, "Model I" is the ResNeXt50 model with SpinalNet FC, "Model II" is the DenseNet169 model with a Standard FC layer, "Model III" is the DenseNet201 model with SpinalNet FC, "Outputs 1–3" are the outputs of these three models, and "Prediction" is the normal/abnormal (fracture) class types. In selection of the three models specified in this ensemble model, the parameters in the training outputs of the 26 CNN models presented in the previous title were taken into consideration. These parameters can be stated as Cohen's kappa score, AUC, and test accuracy. The steps of the procedure and a block diagram in Figure 15 for the proposed EL1 ensemble model are provided as follows:

- Step 1: The last layers of the three pre-trained sub-models in the EL1 ensemble model are adjusted as the Identity layer.
- Step 2: A final layer with 80 outputs for ResNext50 with Spinal FC, 1664 for DenseNet169 with Standard FC, and 960 for DenseNet201 with Spinal FC is achieved.
- Step 3: These outputs are combined to form a single linear layer and a hidden layer with 2704 outputs.
- Step 4: This hidden layer is connected to the classifying layer connected to the sigmoid activation function, the output of which is 2.
- Step 5: The network established as a result of these procedures is re-trained, providing results.

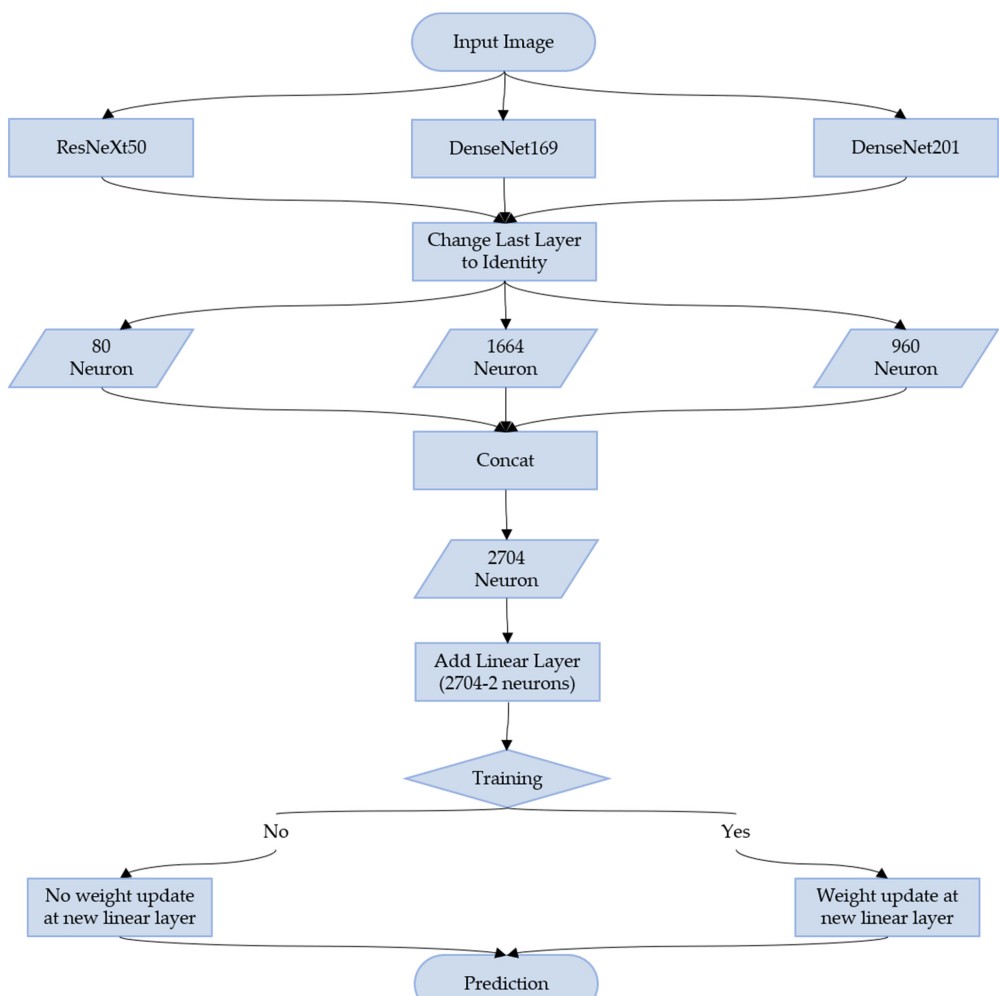

**Figure 15.** A block diagram of the proposed EL1 ensemble model.

### 3.3.2. EL2 (ResNet34 with Spinal FC, DenseNet169 with Standard FC, DenseNet201 with Spinal FC, and ResNext50 with Spinal FC)

The EL2 (Ensemble learning-2) model was developed using a combination of ResNet34 with Spinal FC, DenseNet169 with Standard FC, DenseNet201 with Spinal FC, and a sub-ensemble model. A schematic diagram of the developed model is provided in Figure 16. It is observed upon examination of the figure that the EL2 model, differently from the EL1 model, consists of three single models and one sub-ensemble model and has a different structure. The "X-ray Images" specified in the figure refers to the dataset containing shoulder bone images, "Model I" to the ResNet34 model with Spinal FC, "Model II" to the DenseNet201 model with Spinal FC, "Model III" to the sub-ensemble model, "Model IV" to the DenseNet169 model with Standard FC, and "Predictions I–IV" refer to the outputs of the classification performed with these four models. The parts of the procedure for the proposed EL2 ensemble model are as follows:

- Part 1: In the "Input" section, there is a shoulder X-ray image dataset that has been subjected to certain image processing techniques.
- Part 2: In the "Classification Network" section, the models that establish our ensemble model are defined. There are three single sub-models and a sub-ensemble model that constitute our ensemble model therein. Our sub-ensemble model is an architecture trained by connecting the predicted outputs of ResNet34 with Spinal FC, DenseNet201, ResNeXt50, and DenseNet169 with Standard FC to a linear layer with eight inputs and two outputs. The evaluation of 26 models as single was effective in the selection of these four models.
- Part 3: In the "Prediction" sections, there are normal/abnormal (fracture) class type outputs achieved as a result of the classification performed in the previous section.
- Part 4: In the "Main Check" section, there is a main check mechanism that plays a role in determining the class of the input image. Therein, Models I and II suggest classifying the input image as abnormal in the final classification, while Models III and IV suggest classifying the input image as normal in the final classification. In cases where suggestions are not available, the classification is carried out with the sub-ensemble model. In the selection of the referred models for each class, Confusion Matrix and Recall parameters previously obtained for the 26 CNN models were taken into consideration.
- Part 5: In the "Sub Check" section, there is a supplementary check mechanism under the main check mechanism. The aim here is to use the classification result of a model (Model IV for Class 1 (abnormal) and Model II for Class 0 (normal)) other than the two models referred to in the main check section as a supplement for the final classification process.
- Part 6: In the "Final Prediction" section, the final output determined as a result of the check mechanisms is achieved.

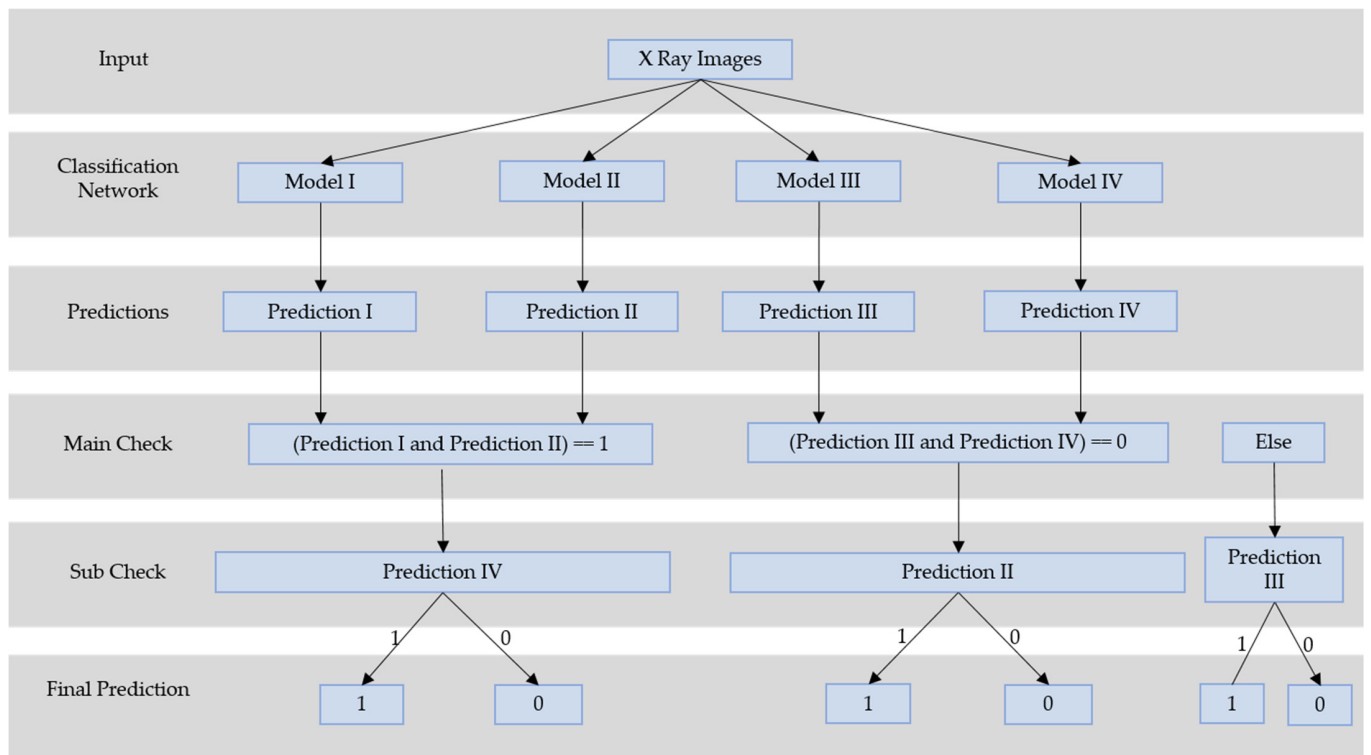

**Figure 16.** The proposed EL2 classification model.

The structure of the EL2 model is explained in Algorithm 1.

---

**Algorithm 1 EL2**

---

**Input:** Shoulder bone X-ray images Dataset = test_dataset
**Process:**
    for image in test_dataset:
        pred_1 = Model_I(image)
        pred_2 = Model_II(image)
        pred_3 = Model_III(image)
        pred_4 = Model_IV(image)
        if(pred_1 and pred_2==1):
            if pred_4==1:
                final_pred = 1:
            if pred_4==0:
                final_pred = 0:
        elif(pred_3 and pred_4==0):
            if pred_2==1:
                final_pred = 1:
            if pred_2==0:
                final_pred = 0:
        else:
            final_pred = pred_3
**Output:** final_pred

---

The block diagram of the proposed EL2 ensemble model is shown in Figure 17.

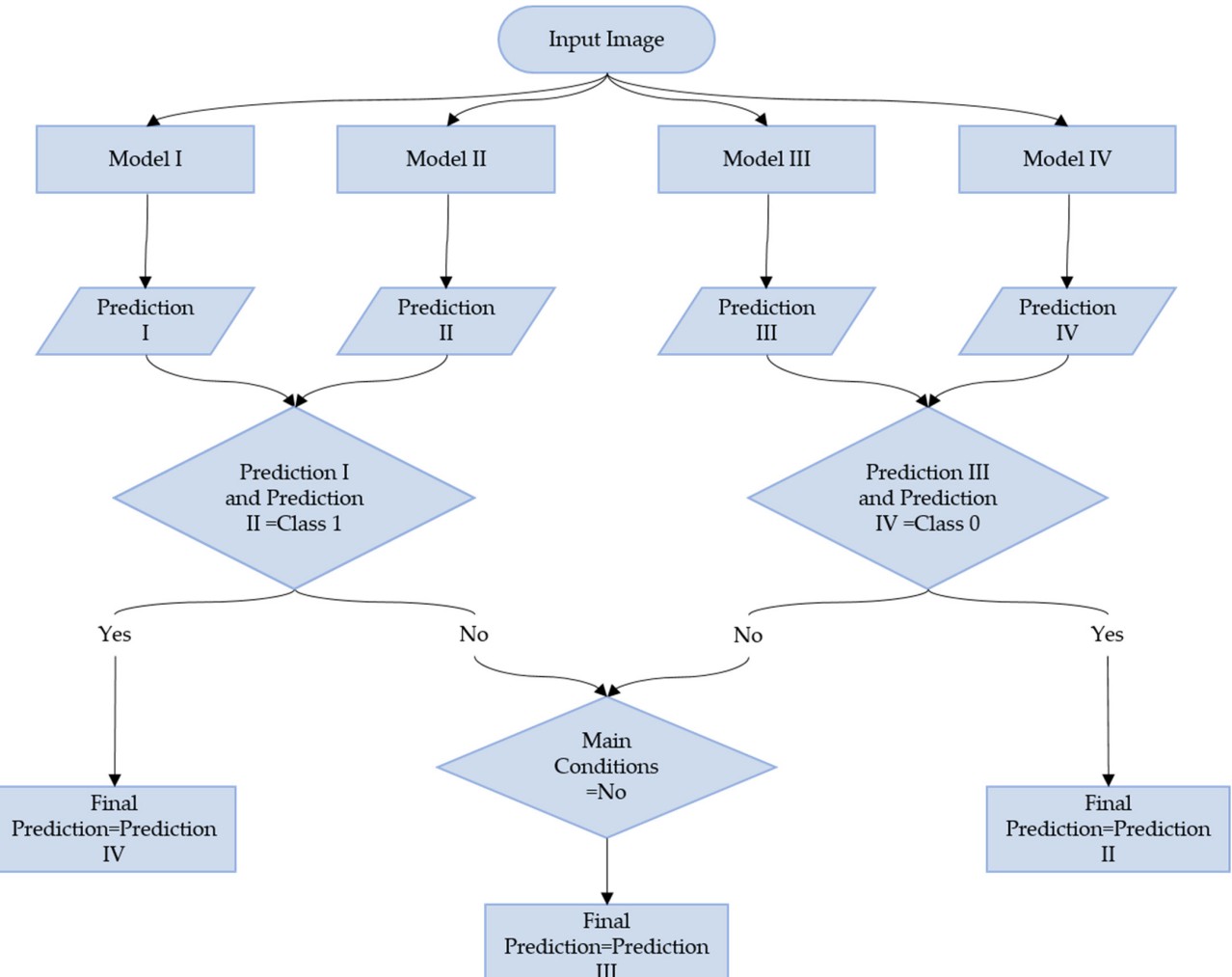

**Figure 17.** A block diagram of the proposed EL2 ensemble model.

## 4. Experiments

In this study, CNN-based deep learning (DL) models were used to classify shoulder bone X-ray images as abnormal (indicating fracture). X-ray images of the shoulder bone in the open source MURA dataset were used. Data pre-processing/augmentation was performed first. Following this, classification was carried out using 13 different CNN models, SpinalNet versions of these models, and two ensemble learning models. The proposed classification models for normal/abnormal (fracture) shoulder bone X-ray images are shown in Figure 18.

### 4.1. Dataset of Shoulder Bone X-ray Images

The MURA dataset is one of the largest among the published open-source radiography datasets. It contains finger, elbow, wrist, hand, forearm, humerus, and shoulder bone X-ray images [2]. In this study, only the shoulder bone X-ray images within the MURA dataset were used mainly because it is the most balanced type in the MURA dataset in terms of distribution of the amount of data provided for both training and validation. This balanced distribution is presented in Figure 19. A balanced dataset can otherwise be obtained with data augmentation or synthetic data generation to avoid problems that may arise when working with an imbalanced dataset. Although the MURA dataset is an open-source dataset, only the training and validation datasets are publicly available. The classification models used in the first study and in studies conducted in competition conducted with

the MURA dataset were tested using test data that are not publicly available. Due to the confidential nature of the test data and the inability to conduct testing with these data as other studies have, the validation data was used as test data in this study.

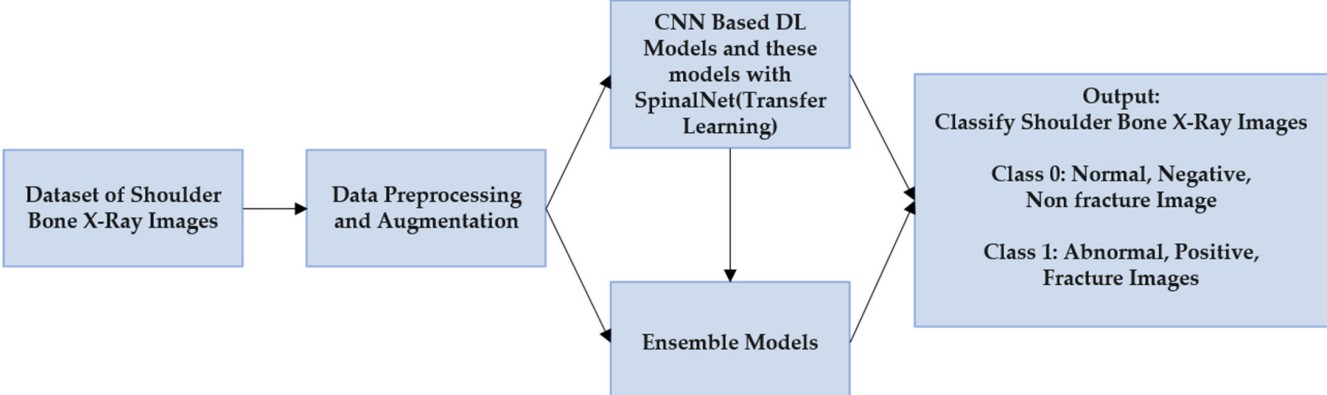

**Figure 18.** The proposed models for classification of shoulder bone X-ray images.

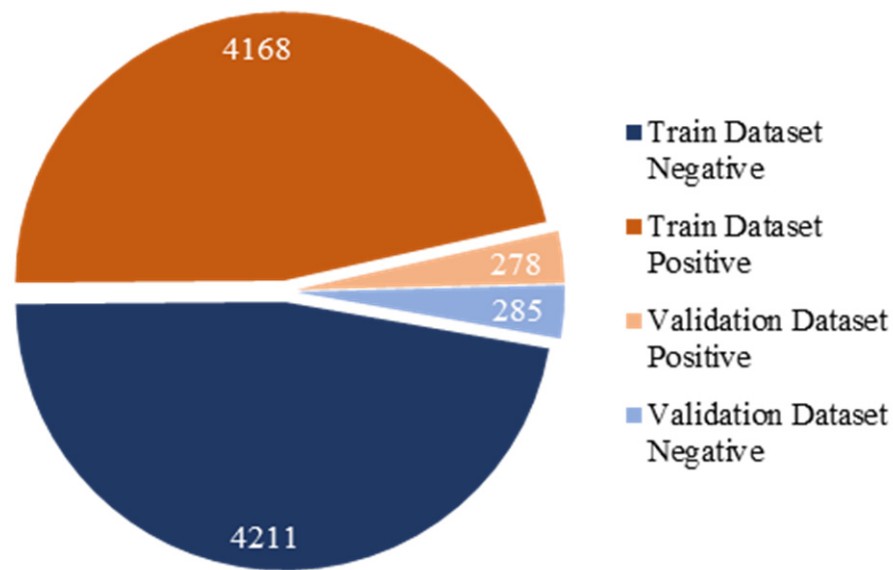

**Figure 19.** The shoulder bone X-ray dataset [2].

These images, which initially had different resolutions and three channels, were first pre-processed and then converted to $320 \times 320 \times 3$ pixels before being used for the deep learning model. The size $320 \times 320 \times 3$ was chosen because it is the resolution most compatible with other studies using this dataset. The data formats are png, and no alterations were made in terms of the format type. The image type of the shoulder bone X-ray images, the number of training images, the number of test images, and the original and new image sizes are provided in the Table 2.

**Table 2.** Details of the shoulder bone X-ray images used in the study [2].

| Shoulder Bone X-ray Images | Image Types | Train Dataset | Test Dataset | Org. Image Size | New Image Size |
|---|---|---|---|---|---|
| Class 0: Normal (Negative) Bone X-ray Images | png, 3-ch. | 4211 | 285 | various | $320 \times 320 \times 3$ |
| Class 1: Abnormal (Positive) Bone X-ray Images | png, 3-ch. | 4168 | 278 | various | $320 \times 320 \times 3$ |
| Total | png, 3-ch. | 8379 | 563 | various | $320 \times 320 \times 3$ |

The table above shows that the image sizes were originally different but later converted to $320 \times 320 \times 3$ in size. All images are in png format. The original quantity of normal images was 4211 in the training section and 285 in the test section, and that of abnormal images was 4168 in the training section and 278 in the test section.

*4.2. Data Augmentation for Shoulder Bone X-ray Images*

Since the results obtained in deep learning, particularly in classification studies, are very sensitive to the dataset, and since an increase in the amount of data in the training stage of the network has a positive effect on the training of the network, the quantity of shoulder bone X-ray images in the MURA dataset used in this study was increased by data augmentation. For this procedure, new images were obtained by rotating the images to the right or to the left to a maximum of 10 degrees.

*4.3. Data Pre-Processing for Shoulder Bone X-ray Images*

There is both noise and a dark background in the images observed in this study, which may adversely affect the classification and/or fracture detection processes. Various pre-processing steps were applied to the dataset to eliminate the abovementioned adverse effects as much as possible and to obtain more accurate results in the classification process. These pre-processing steps are listed below:

- Detection of the Corresponding Area: Most of the X-ray images in the used dataset were insufficient in terms of semantic information in relation to the image size. In order to eliminate such insufficiency, the images were first converted to gray-scale and then subjected to double thresholding and to an adaptive threshold value determined using Otsu's thresholding value method. In the gray-scale images, the within-class variance value corresponding to all possible threshold values for the two color classes assumed as background and foreground was calculated. The threshold value that made this variance the smallest was the optimal threshold value. This method is known as Otsu's thresholding value method [35]. Subsequently, the edge in the thresholded image was determined using edge detection methods. After this process, the original image was cropped based on the calculated values.
- CLAHE Transformation: In the next step, the contrast-limited adaptive histogram equalization (CLAHE) transformation in the OpenCV library was used. In the transformation used, the input image was divided into parts by the user, with each part containing a histogram within itself. The histogram of each part was then adjusted based on the histogram cropping limit entered by the user, and all parts were finally brought together to obtain a Clahe-transformed version of the input image [36,37]. New outputs were achieved by contrast equalization of the cropped images by the CLAHE method.
- Normalization and Standardization: In the last step, the images were normalized and standardized using the image-net values.

For each class (normal/abnormal (fracture)), an original shoulder bone X-ray image and an image subjected to the abovementioned pre-processing steps are provided as examples in Figure 20.

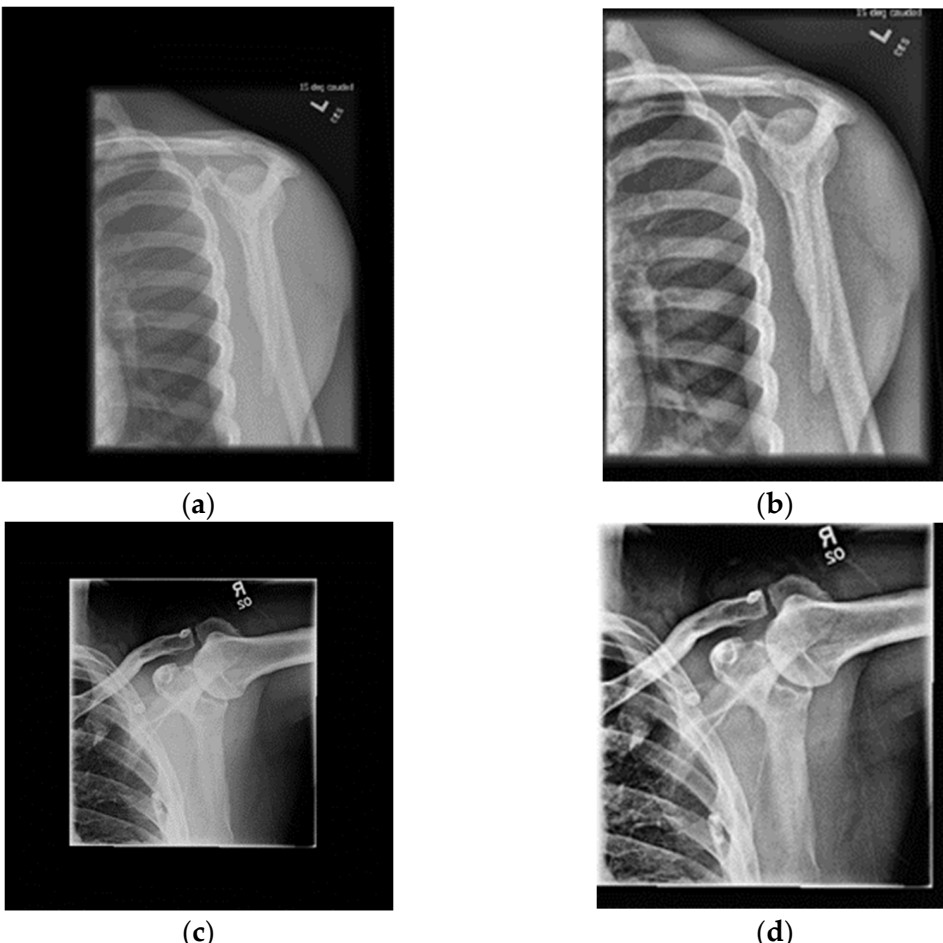

**Figure 20.** Original and pre-processed X-ray images of normal/abnormal (fractured) shoulder bone. (**a**) original normal; (**b**) pre-processed normal; (**c**) original abnormal; (**d**) pre-processed abnormal [2].

In the figure above, section a shows an original version of a normal (negative) shoulder bone X-ray image; section b shows a cropped and CLAHE pre-processed version of this normal image; section c shows an original version of an abnormal (positive, fractured) shoulder bone X-ray image; section d shows a cropped and CLAHE pre-processed version of this abnormal image.

### 4.4. Classification Results

Online servers were used as hardware in the process of classification of shoulder bone X-ray images in this study. All classification processes were carried out on Google Colab with nVIDIA Tesla T4 16GB GDDR6 graphics cards. In addition, the torchvision, seaborn, scikitlearn, matplotlib, and torch packages were used. The program codes written using the PyTorch deep learning library are publicly available at https://github.com/fatihuysal88/shoulder-c (accessed on 1 March 2021). A learning rate of 0.0001, an epoch number of 40, the Adam optimizer, and the cross-entropy loss function were used in all classification procedures performed with the ResNet, ResNeXt, DenseNet, VGG, InceptionV3, and MobileNetV2 deep learning models. While the same learning rate, epoch number, optimizer, and loss function were used in the versions with SpinalNet of these models, Spinal FC was used in the classification layer. The learning rate value initially used with these models

was not fixed, and it was decreased 10 times every 10 epochs to increase network learning success. The same parameters were used in the ensemble learning models.

### 4.4.1. Evaluation Metrics

In order to evaluate the results obtained in classification problems accurately and completely, the following must be obtained for each class: accuracy percentage, confusion matrix, precision, recall, F1-score from each classification model, ROC curves, and AUC scores. The accuracy percentage refers to what percentage of test data was correctly classified. The confusion matrix provides a table of the current status in the dataset and the number of correct and incorrect predictions in our classification model. This table contains true positive (TP), true negative (TN), false positive (FP), and false negative (FN) values. Precision refers to values predicted as positive. Recall is defined as the true positive rate. F1-score is the harmonic mean of precision and recall values. Cohen's kappa coefficient is a statistical method used to measure the reliability of agreement between two raters. The $p_0$ value used in this calculation refers to the result of accuracy, and the $p_e$ value refers to the probability of random agreement. The following Tables 3–6 contain information on what TP, TN, FP, FN, and the confusion matrix refer to and how the accuracy, precision, recall, F1-score, Cohen's-kappa score, ROC curve, and AUC scores are calculated.

**Table 3.** TP, TN, FP, and FN calculations.

|  | Label Value | Prediction Value |
| --- | --- | --- |
| **TP** | positive | positive |
| **TN** | negative | negative |
| **FP** | positive | negative |
| **FN** | negative | positive |

**Table 4.** Confusion matrix.

|  | Predicted Class Negative | Predicted Class Positive |
| --- | --- | --- |
| **Actual Class: Negative** | TN | FP |
| **Actual Class: Positive** | FN | TP |

**Table 5.** Calculation of Cohen's kappa score parameters.

| $\mathbf{P_0}$ | (TP + TN)/(TP + TN + FP + FN) |
| --- | --- |
| $\mathbf{P_{positive}}$ | $(TP + FP)(TP + FN)/(TP + TN + FP + FN)^2$ |
| $\mathbf{P_{negative}}$ | $(FN + TN)(FP + TN)/(TP + TN + FP + FN)^2$ |
| $\mathbf{P_e}$ | $P_{positive} + P_{negative}$ |

**Table 6.** Evaluation metrics.

| **Confusion Matrix** | TP, TN, FP, FN |
| --- | --- |
| **Training/Testing Accuracy** | (TP + TN)/(TP + TN + FP + FN) |
| **Precision** | TP/(TP + FP) |
| **Recall** | TP/(TP + FN) |
| **F1-score** | 2TP/(2TP + FP + FN) |
| **Cohen's kappa score** | $(p_0 - p_e)/(1 - p_e)$ |
| **ROC curve** | TP rate—FP rate change |
| **AUC scores** | Area under the ROC curve |

### 4.4.2. Classification Results of 13 CNN-Based Deep Learning Models with Standard FC/Spinal FC

The training accuracy, test accuracy, precision, recall, F1-score, and Cohen's kappa scores obtained as the result of the classification performed with each model are presented in the following figures and Tables 7–12.

**Table 7.** Training accuracy results of classification models.

| Models | Standart FC | Spinal Net | Models | Standart FC | Spinal Net |
|---|---|---|---|---|---|
| DenseNet169 | 0.8883 | 0.8666 | ResNet101 | 0.8419 | 0.8225 |
| DenseNet201 | 0.8934 | 0.8694 | ResNet152 | 0.845 | 0.8346 |
| InceptionV3 | 0.8882 | 0.8914 | ResNeXt50 | 0.8707 | 0.8654 |
| MobileNetV2 | 0.8647 | 0.8845 | ResNeXt101 | 0.8539 | 0.8303 |
| ResNet34 | 0.8673 | 0.8517 | VGG13 | **0.9389** | 0.8507 |
| ResNet50 | 0.8489 | 0.8395 | VGG16 | 0.8698 | 0.812 |
| | | | VGG19 | 0.9055 | 0.8011 |

**Table 8.** Test accuracy results of classification models.

| Models | Standart FC | Spinal Net | Models | Standart FC | Spinal Net |
|---|---|---|---|---|---|
| DenseNet169 | **0.8419** | 0.8152 | ResNet101 | 0.817 | 0.8188 |
| DenseNet201 | 0.8206 | 0.8294 | ResNet152 | 0.8117 | 0.817 |
| InceptionV3 | 0.8259 | 0.817 | ResNeXt50 | 0.817 | 0.8241 |
| MobileNetV2 | 0.8241 | 0.8099 | ResNeXt101 | 0.8206 | 0.8082 |
| ResNet34 | 0.8188 | 0.8206 | VGG13 | 0.7797 | 0.8223 |
| ResNet50 | 0.8188 | 0.8081 | VGG16 | 0.785 | 0.801 |
| | | | VGG19 | 0.785 | 0.8046 |

**Table 9.** Precision results of classification models.

| Models | Standart FC | Spinal Net | Models | Standart FC | Spinal Net |
|---|---|---|---|---|---|
| DenseNet169 | **0.845** | 0.815 | ResNet101 | 0.815 | 0.82 |
| DenseNet201 | 0.82 | 0.83 | ResNet152 | 0.815 | 0.82 |
| InceptionV3 | 0.825 | 0.82 | ResNeXt50 | 0.82 | 0.825 |
| MobileNetV2 | 0.83 | 0.81 | ResNeXt101 | 0.82 | 0.815 |
| ResNet34 | 0.825 | 0.82 | VGG13 | 0.8 | 0.825 |
| ResNet50 | 0.815 | 0.81 | VGG16 | 0.79 | 0.805 |
| | | | VGG19 | 0.785 | 0.81 |

**Table 10.** Recall results of classification models.

| Models | Standart FC | Spinal Net | Models | Standart FC | Spinal Net |
|---|---|---|---|---|---|
| DenseNet169 | **0.84** | 0.815 | ResNet101 | 0.815 | 0.82 |
| DenseNet201 | 0.815 | 0.83 | ResNet152 | 0.81 | 0.815 |
| InceptionV3 | 0.825 | 0.815 | ResNeXt50 | 0.815 | 0.825 |
| MobileNetV2 | 0.82 | 0.81 | ResNeXt101 | 0.815 | 0.815 |
| ResNet34 | 0.815 | 0.82 | VGG13 | 0.78 | 0.825 |
| ResNet50 | 0.82 | 0.81 | VGG16 | 0.785 | 0.8 |
| | | | VGG19 | 0.785 | 0.805 |

**Table 11.** F1-score results of classification models.

| Models | Standart FC | Spinal Net | Models | Standart FC | Spinal Net |
|---|---|---|---|---|---|
| DenseNet169 | **0.84** | 0.815 | ResNet101 | 0.82 | 0.82 |
| DenseNet201 | 0.82 | 0.83 | ResNet152 | 0.81 | 0.815 |
| InceptionV3 | 0.825 | 0.815 | ResNeXt50 | 0.815 | 0.825 |
| MobileNetV2 | 0.82 | 0.81 | ResNeXt101 | 0.82 | 0.815 |
| ResNet34 | 0.82 | 0.82 | VGG13 | 0.775 | 0.82 |
| ResNet50 | 0.815 | 0.805 | VGG16 | 0.785 | 0.8 |
| | | | VGG19 | 0.785 | 0.805 |

**Table 12.** Cohen's kappa results of classification models.

| Models | Standart FC | Spinal Net | Models | Standart FC | Spinal Net |
|--------|-------------|------------|--------|-------------|------------|
| DenseNet169 | **0.6834** | 0.6302 | ResNet101 | 0.634 | 0.6372 |
| DenseNet201 | 0.641 | 0.6588 | ResNet152 | 0.6231 | 0.6332 |
| InceptionV3 | 0.6514 | 0.6338 | ResNeXt50 | 0.6338 | 0.648 |
| MobileNetV2 | 0.6478 | 0.6195 | ResNeXt101 | 0.641 | 0.6267 |
| ResNet34 | 0.6372 | 0.6411 | VGG13 | 0.558 | 0.6442 |
| ResNet50 | 0.6375 | 0.6161 | VGG16 | 0.5695 | 0.6014 |
|  |  |  | VGG19 | 0.5698 | 0.6085 |

Figure 21 shows that the highest training accuracy in classification is achieved by VGG13 among models with Standard FC and by InceptionV3 among models with Spinal FC. The test accuracy in Figure 22, the precision in Figure 23, the recall in Figure 24, the F1-score in Figure 25, and Cohen's kappa score in Figure 26 show that the highest scores are achieved by DenseNet169 among models with Standard FC and by DenseNet201 among models with Spinal FC.

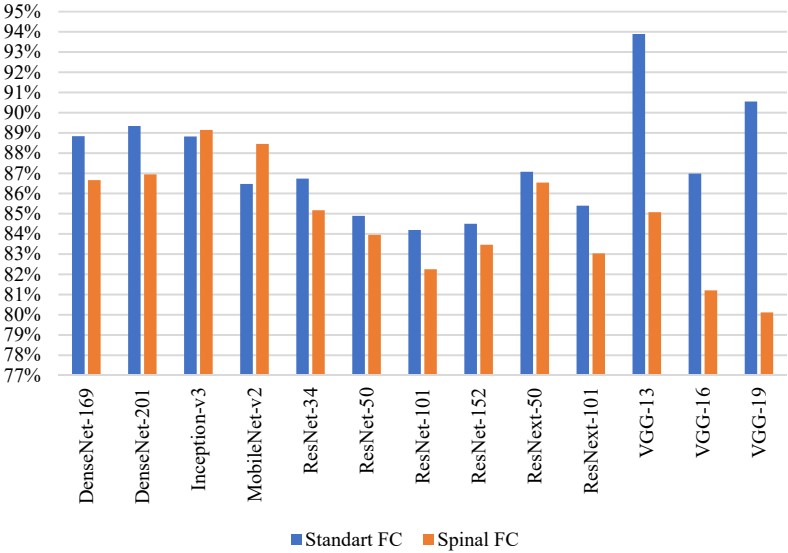

**Figure 21.** Training accuracy results of classification models.

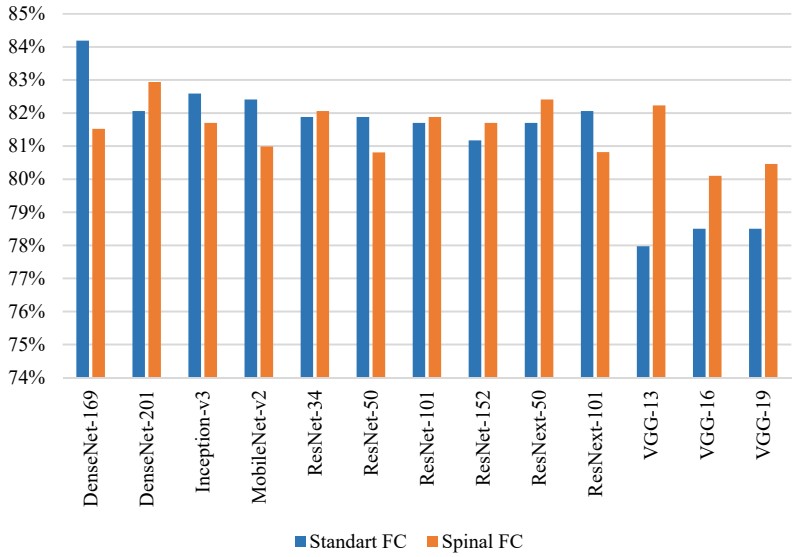

**Figure 22.** Test accuracy results of classification models.

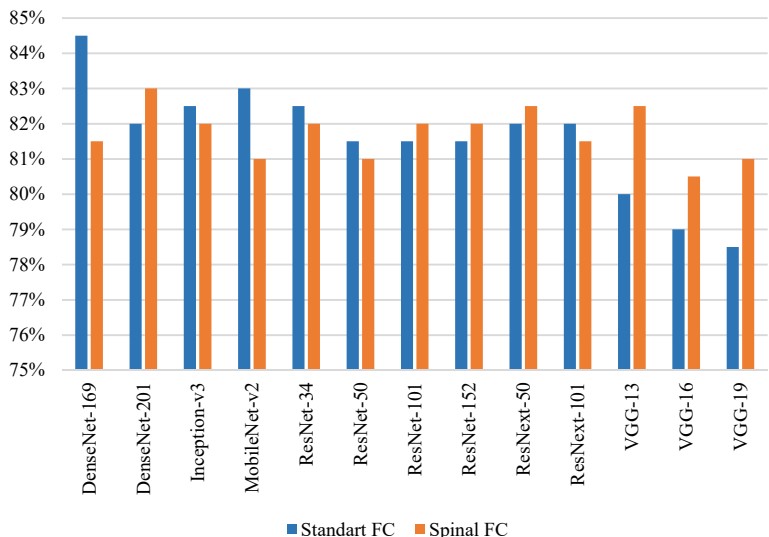

**Figure 23.** Precision results of classification models.

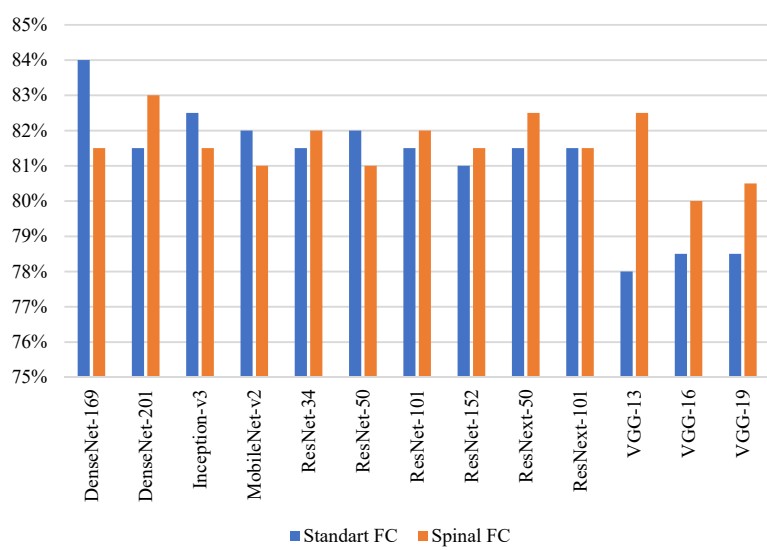

**Figure 24.** Recall results of classification models.

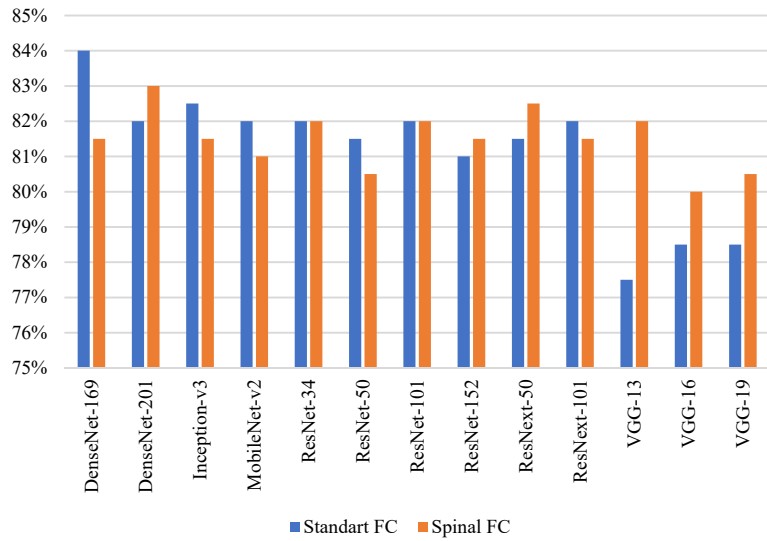

**Figure 25.** F1-score results of classification models.

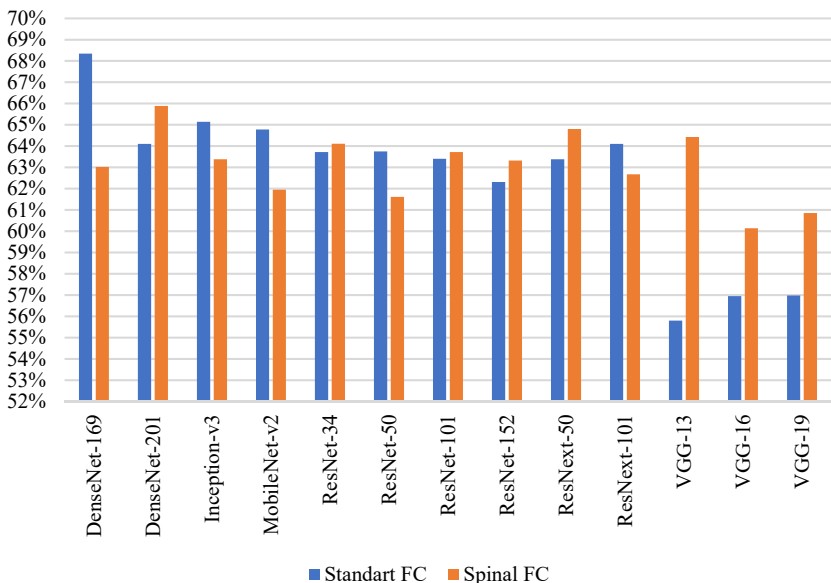

**Figure 26.** Cohen's kappa results of classification models.

The confusion matrix values and AUC scores achieved as a result of 26 classification procedures, performed with DenseNet, Inception, MobileNet, ResNet, ResNeXt, and VGG models with Standard FC and Spinal FC with different numbers of layers, are provided in Table 13.

**Table 13.** Confusion matrix values and AUC scores of 26 classification models (a: with Standard FC, b: with Spinal FC).

| Models | TP | FP | FN | TN | Class0: AUC | Class1: AUC |
|---|---|---|---|---|---|---|
| DenseNet169a | 222 | 33 | 56 | 252 | **0.8809** | **0.8797** |
| DenseNet169b | 218 | 44 | 60 | 241 | 0.8602 | 0.8598 |
| DenseNet201a | 225 | 48 | 53 | 237 | 0.8584 | 0.8653 |
| DenseNet201b | 228 | 46 | 50 | 239 | 0.8727 | 0.8724 |
| InceptionV3a | 218 | 38 | 60 | 247 | 0.8754 | 0.8707 |
| InceptionV3b | 220 | 45 | 58 | 240 | 0.8582 | 0.8585 |
| MobileNetV2a | 215 | 36 | 63 | 249 | 0.8777 | 0.8378 |
| MobileNetV2b | 216 | 45 | 62 | 240 | 0.8633 | 0.861 |
| ResNet34a | 215 | 39 | 63 | 246 | 0.8705 | 0.8767 |
| ResNet34b | 228 | 51 | 50 | 234 | 0.8617 | 0.8619 |
| ResNet50a | 224 | 48 | 54 | 237 | 0.8715 | 0.8662 |
| ResNet50b | 219 | 49 | 59 | 236 | 0.8588 | 0.8584 |
| ResNet101a | 228 | 53 | 50 | 232 | 0.8683 | 0.8703 |
| ResNet101b | 216 | 40 | 62 | 245 | 0.8609 | 0.861 |
| ResNet152a | 217 | 45 | 61 | 240 | 0.8648 | 0.8701 |
| ResNet152b | 220 | 45 | 58 | 240 | 0.8597 | 0.8606 |
| ResNeXt50a | 221 | 46 | 57 | 239 | 0.8644 | 0.8699 |
| ResNeXt50b | 223 | 44 | 55 | 241 | 0.8789 | 0.8783 |
| ResNeXt101a | 225 | 48 | 53 | 237 | 0.8772 | 0.8765 |
| ResNeXt101b | 219 | 46 | 59 | 239 | 0.8652 | 0.8561 |
| VGG13a | 181 | 27 | 97 | **258** | 0.8406 | 0.8415 |
| VGG13b | **231** | 35 | 65 | 250 | 0.8705 | 0.8737 |
| VGG16a | 203 | 46 | 75 | 239 | 0.8517 | 0.8523 |
| VGG16b | 204 | 38 | 74 | 247 | 0.8542 | 0.857 |
| VGG19a | 212 | 55 | 66 | 230 | 0.8374 | 0.8502 |
| VGG19b | 205 | 37 | 73 | 248 | 0.858 | 0.8539 |

The table shows that the highest classification accuracy (TP value) out of the 278 Class 1 (positive, abnormal, fracture) images in the test data is achieved by the VGG13 model with Spinal FC, with 231 images. Moreover, out of 285 images in Class 0 (negative, normal) in the test data, the highest classification accuracy (TN value) was achieved by the VGG13 model with Standard FC, with 258 images. The highest AUC score in Class 1 is 0.8797, and the highest score in Class 0 is 0.8809, and both were achieved by the DenseNet169 model with Standard FC.

In addition to the 26 classification models that were used for the classification of shoulder bone X-ray images, two different ensemble models were developed, which further improved the classification results.

### 4.4.3. Classification Results of Our Ensemble Models

ResNeXt50 with Spinal FC, DenseNet169 with Standard FC, and DenseNet201 with Spinal FC were used in EL1:

- ResNeXt50 with Spinal FC was selected because the AUC score achieved by detecting fracture images is the second highest, with 0.8783, after DenseNet169 with Standard FC.
- DenseNet169 with Standard FC was selected because its classification had the highest test accuracy, Cohen's kappa score, and AUC score among all models used in this study.
- DenseNet201 with Spinal FC was selected because the test accuracy and Cohen's kappa score were the second highest after DenseNet169 with Standard FC among all models and the highest among models with Spinal FC.

The classification results of the EL1 model are provided in the following figures and tables. In Table 14, ResNeXt50b represents the ResNeXt50 model with Spinal FC, DenseNet169a represents the DenseNet169 model with Standard FC, and DenseNet201b represents the DenseNet201 model with Spinal FC.

**Table 14.** Classification results of EL1.

| Models | Test Acc. | Pre. | Recall | F1-Score | Cohen's Kappa |
|--------|-----------|------|--------|----------|---------------|
| ResNeXt50b | 0.8241 | 0.825 | 0.825 | 0.825 | 0.648 |
| DenseNet169a | 0.8419 | 0.845 | **0.84** | 0.84 | 0.6834 |
| DenseNet201b | 0.8294 | 0.83 | 0.83 | 0.83 | 0.6588 |
| EL1 | **0.8455** | **0.8631** | 0.8165 | **0.8455** | **0.6907** |

The table above shows that, with the EL1 model, the test accuracy is 0.8455 and Cohen's kappa score is 0.6907. Therefore, the highest classification result obtained with DenseNet169 with Standard FC given in the previous section was exceeded.

The confusion matrix specified in Figure 27 was a result of the classification performed using the EL1 model. The following results are concluded as per the figure: TN: 249, FP: 36, FN: 51, and TP: 227.

The ROC curve representing the change in the TP rate and FP rate of the EL1 model entails an AUC score of 0.8862 for Class 0 (normal) and Class 1 (abnormal, fracture) in Figure 28. This AUC score is also higher than the AUC achieved with ResNet169 with Spinal FC in the previous section.

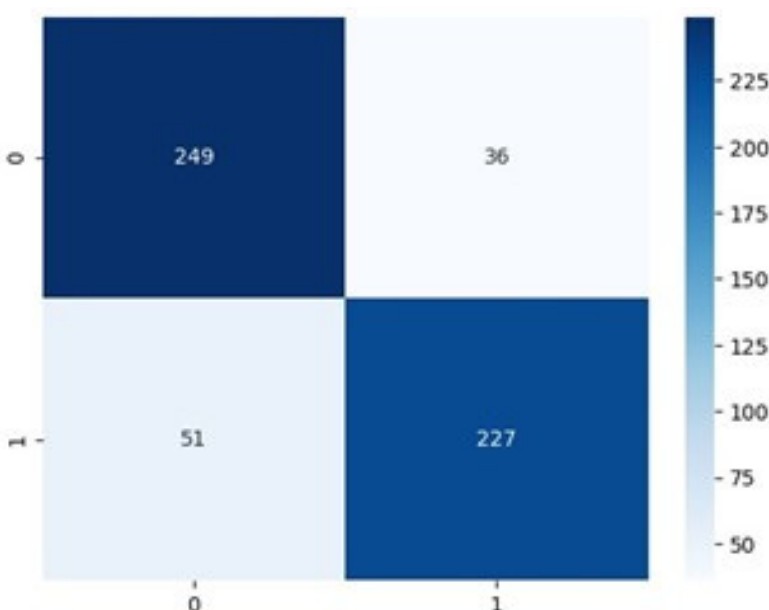

**Figure 27.** Confusion matrix results of EL1.

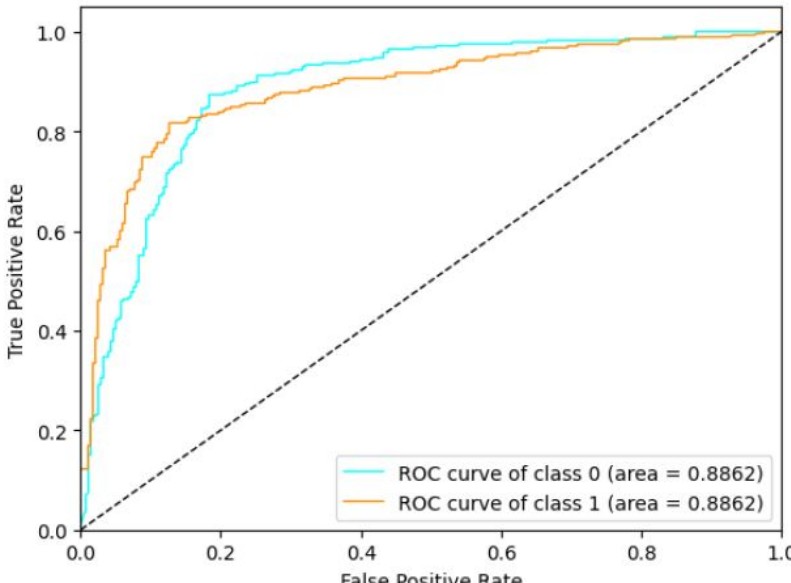

**Figure 28.** ROC curves and AUC results of EL1.

Three different single classification models and a sub-ensemble model were used in the EL2 model: ResNet34 with Spinal FC, DenseNet201, DenseNet169 with Standard FC, and sub-ensemble models. The TP and TN values in the confusion matrix scores and recall scores from the 26 different classification models used in the first stage were taken into consideration when choosing the three single models used in the EL2 model and developing the sub-ensemble model. The results of the classification carried out with the EL2 model are provided in Table 15, where ResNet34b represents the ResNet34 model with Spinal FC, DenseNet169a represents the DenseNet169 model with Standard FC, DenseNet201b represents the DenseNet201 model with Spinal FC, and SubEnsemble represents the developed sub-ensemble model.

**Table 15.** Classification results of EL2.

| Models | Test Acc. | Pre. | Recall | F1-score | Cohen's Kappa |
|---|---|---|---|---|---|
| ResNet34b | 0.8206 | 0.82 | 0.82 | 0.82 | 0.6411 |
| DenseNet169a | 0.8419 | 0.815 | 0.84 | 0.84 | 0.6834 |
| DenseNet201b | 0.8294 | 0.83 | 0.83 | 0.83 | 0.6588 |
| SubEnsemble | 0.8401 | 0.84 | 0.84 | 0.84 | 0.6799 |
| EL2 | **0.8472** | **0.85** | **0.845** | **0.845** | **0.6942** |

The table shows that Cohen's kappa is 0.6942, and the test accuracy score is 0.8472. Therefore, the EL2 model outperforms the 26 CNN-based models and the EL1 model.

Figure 29 presents the confusion matrix scores achieved as a result of the classification carried out with the EL2 model: TN: 248, FP: 36, FN: 49 and TP: 229. For comparison, the TP in the EL1 model was 227.

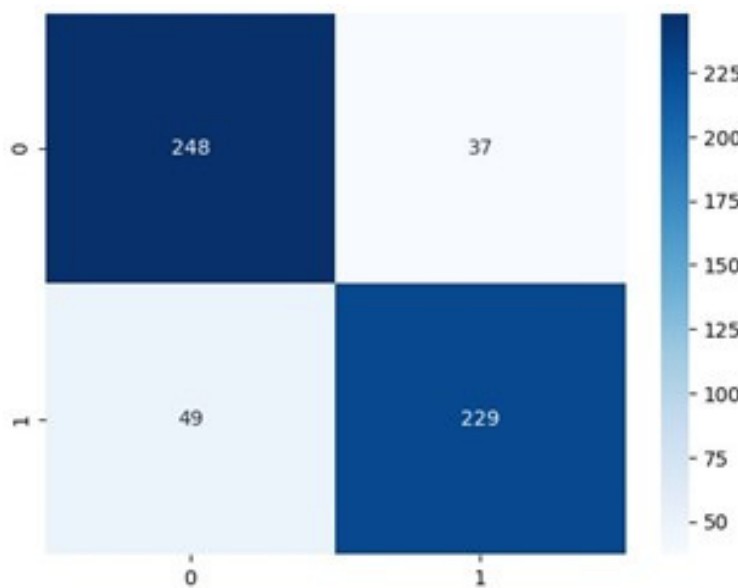

**Figure 29.** Confusion matrix results of EL2.

The ROC curves and AUC scores achieved as a result of the classification performed with the EL2 model are provided in Figure 30. The AUC scores are 0.8698 for Class 0 (normal, negative) and 0.8695 for Class 1 (abnormal, positive, fracture). Although these AUC scores appear to be slightly decreased compared to the EL1 model, since the test accuracy, Cohen's kappa scores, and the number of detected shoulder bone fracture images (TP) in the EL2 model are higher than the EL1 model, the EL2 model overall provides better results.

The test accuracy values obtained in classification studies vary depending on the amount and type of test data. In other classification studies on shoulder images in the MURA dataset, different approaches are used to obtain test data. Therefore, the comparison of classification results of the 28 models used in this study will be more appropriate. Comparison with the classification results obtained in other studies may be misleading due to the differences in the test data. Considering this, Table 16 of other important shoulder bone classification studies available in the literature, regarding the dataset, amount, method used, and classification accuracies.

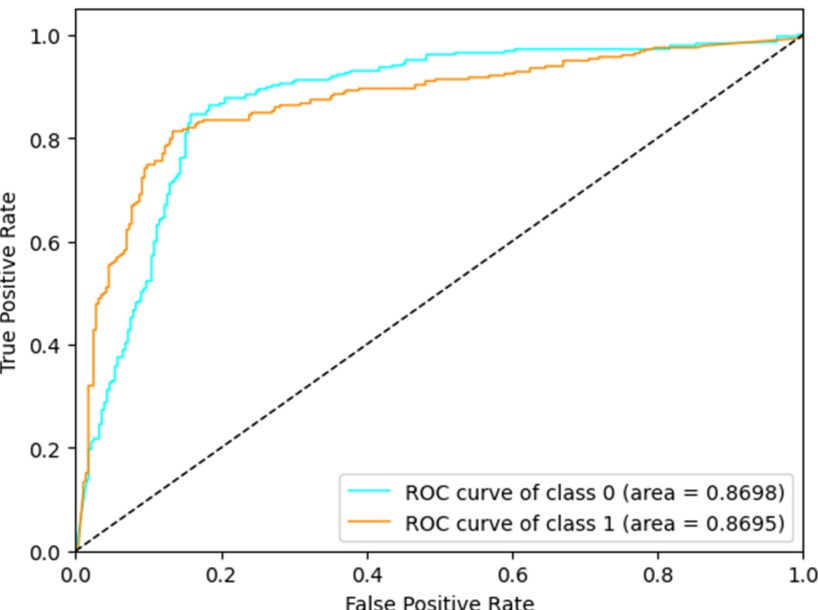

**Figure 30.** ROC curves and AUC results of EL2.

**Table 16.** Comparison with previous studies.

| Studies | Dataset | Best Method | Classification Type and Test Accuracy |
|---|---|---|---|
| **Our study** | **MURA Shoulder, 563 validation/test images, open data** | **EL2** | **Fracture/normal: 0.8472** |
| Liang and Gu [5] | MURA dataset, 194 validation images | GCN | Fracture/normal: 0.9112 |
| Saif et al. [6] | MURA dataset, various test images, %50 test data, test quantity not specified | Capsule Network | Fracture/normal: 0.9208 |
| Chung et al. [15] | 1376 CT images | ResNet152 | Normal/four different type: 0.96 |
| Urban et al. [17] | 597 X-ray images | NASNet | Implant: 0.804 |
| Sezers [16] | 219 MR images | CNN | Normal/edematous/Hill–Sachs lesions: 0.9843 |
| Sezers [18] | 219 MR images | Extreme learning machines | Normal/edematous/Hill–Sachs lesions: 0.94 |
| Sezers [19] | 1006 MR images | CapsNet | Normal/degenerated/torn: 0.9474 |

## 5. Conclusions and Future Work

The aim of this study was to find the most optimal model for classifying shoulder bone X-ray images as normal or abnormal (fracture). In order to achieve this aim, the first objective was to classify fracture images using 26 classification models, i.e., Standard FC and Spinal FC versions of 13 different CNN-based models. Based on the results obtained therein, two different ensemble models (EL1 and EL2) were developed and used for classification procedures in order to further increase classification accuracy and Cohen's kappa score. Among the 28 different classifications models, the highest test accuracy and Cohen's kappa score were achieved in the EL2 model, and the highest AUC score was achieved in the EL1 model. In the EL1 model, the fully connected layers of the models trained on the dataset were combined and turned into a single fully connected layer. This new layer was linked to the classification layer. In this way, in the last training part performed in the EL1 model, the weights in the new fully connected layer were updated, and the best configuration of the three models was obtained. For this reason, the EL1 model can be used on different datasets because it updates the weights of the new FC layer for the best configuration on any dataset. In the EL2 model, the sub-models consulted in the decision mechanism consist of models that show the best performance on the dataset. For this reason, if one is working on their own dataset, they should determine their sub-models

according to the performance results obtained with transfer learning. The reason for using transfer learning is the further improvement of classifications made with models that have not been pre-trained. With this approach, a serious increase in the classification problem was observed. Ensemble learning was mainly used to further improve the classification results obtained with 26 different deep learning models and to bring new approaches to the literature in this field. The contribution of the study to the literature is as follows.

- In similar studies, binary classification is mostly performed. However, while there are mainly two classes (normal/abnormal) in this study, differently from the literature, multi-class classification was carried out, in order to determine the most compatible models to be used in ensemble models, developed by evaluating the outputs of each class of the 26 classification models initially used. This allowed the best results in this study to be achieved with ensemble models.
- This is the first time that Spinal FC, which, compared to the Standard FC, has a lower number of weights in the hidden layer, was used in many models (Inception, ResNeXt, and MobileNet). Moreover, SpinalNet was used on medical images for the first time, and it had a positive effect on more than half of the classification results.
- A unique structure is introduced, since the reliability of the detection of classes was used as a basis when designing the EL2 model, which further improves classification results.

The aim of this study, focused on the detection of shoulder bone fractures in X-ray images, is to help physicians diagnose fractures in the shoulder and apply the required treatment. Following this study, a real-time mobile application that detects fractures and/or fracture areas in shoulder bones should be developed, specifically to help physicians performing emergency services.

**Author Contributions:** Conceptualization, F.U., F.H., and O.P.; methodology, F.U., F.H., and O.P.; software, F.U., F.H., and O.P.; validation, F.U., F.H., and O.P.; formal analysis, F.U., F.H., and O.P.; investigation, F.U., F.H., and O.P.; resources, F.U., F.H., and O.P.; data curation, F.U., F.H., and O.P.; writing—original draft preparation, F.U., F.H., and O.P.; writing—review and editing, F.U., F.H., O.P., T.T., and N.T.; visualization, F.U., F.H., and O.P.; supervision, F.U., F.H., and O.P. All authors have read and agreed to the published version of the manuscript.

**Funding:** This research received no external funding.

**Institutional Review Board Statement:** Not applicable.

**Informed Consent Statement:** Not applicable.

**Data Availability Statement:** Data used in this study are available at https://stanfordmlgroup.github.io/competitions/mura (accessed on 1 September 2020).

**Conflicts of Interest:** The authors declare no conflict of interest.

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
