# Peer review of "Classification of Shoulder X-ray Images with Deep Learning Ensemble Models"

_applsci, doi:10.3390/app11062723_

Round 1

Reviewer 1 Report

This paper proposes a simple ensemble technique to combine results of existing neural network models for improved prediction. The contribution is incremental and the quality of the paper should be significantly improved before being considered for publication. My major concerns include:

  1. How is the the generalization of the proposed models?
  2. There is very few explanations on why the proposed EL1 and EL2 should be effective. Without any analysis in this regard, it's not convincing that the models could work well on other data sets.
  3. The English writing of the paper needs improvement. Some of the sentences are a bit confusing, e.g., In Abstract, "In EL1 and EL2 models developed using pre-21 trained models with the highest test performance"... In Conclusion, "most optimum model" sounds awkward, please rephrase.
  4. In Introduction, related deep learning models designed for medical image classification should be reviewed, e.g., "Multi-instance deep learning with graph convolutional neural networks for diagnosis of kidney diseases using ultrasound imaging" in MICCAI CLIP'19.
  5. Some parts of the title might be redundant.

Reviewer 2 Report

The topic is a very interesting application however, the paper requires a revision to solve the following issues:

-Please check paper writing, authors should ask a native English-speaker to check the paper.

- A would have also appreciated showing what features have the highest importance.

- It would be very interesting to use other machine learning algorithms comparing their performance, for example using Random Forests or other ensemble methods.

- please revise section 3, and reduce the number of figures by removing unnecessary ones such as fig.15

Reviewer 3 Report

Fractures occur in the shoulder area, which has a wider range of motion than other joints in the body, due to various reasons. To diagnose these fractures the data gathered from X-radiation (X-ray), magnetic resonance imaging (MRI), or computed tomography (CT) are used. This study, is aimed to help physicians by classifying the shoulder images taken from X-Ray devices as fracture / non-fracture with artificial intelligence. For this purpose, the performances of 26 deep learning-based pre-trained models in the detection of shoulder fractures were evaluated on the musculo-18 skeletal radiographs (MURA) dataset, and 2 ensemble learning models (EL1, EL2) were developed. The paper is interesting overall, but following are the comments that must be addressed:

 Comments:

  • In my opinion, the Title of the paper is not suitable Your title is too long and should be reduced. A good title should not exceed 10 words. The title should be clear and informative and should reflect the aim and approach of the work.

Recommendations for titles:

Fewest possible words that describe the contents of the paper.

Avoid waste words like "Studies on", or "Investigations on”, “effects of”, “comparison of”, or “a case of”.

Use specific terms rather than general.

Watch your word order and syntax.

  • The major contribution of the paper looks very weak authors need to explain and clear it ist in at end of the introduction the major contribution.
  • The authors need to draw the block diagram of the proposed approach step by step.
  • Sometimes data usage will be imbalanced in how to tackle imbalanced data to train deep learning algorithms.
  • Before Conclusion, please draw a Table and compare with previous researchers, how your approach is better in terms of accuracy.
  • The authors need to explain why he used the ensemble and TL approach for Fracture and Normal Shoulder Bone X-Ray Images.
  • Authors missing experiment setup??
  • Related work and introduction section need to updated with updated CAD, AI, ML, DL papers such as.

Urban, Gregor, Saman Porhemmat, Maya Stark, Brian Feeley, Kazunori Okada, and Pierre Baldi. "Classifying shoulder implants in X-ray images using deep learning." Computational and structural biotechnology journal 18 (2020): 967-972. Khan, M. A., & Kim, Y. (2021). Cardiac arrhythmia disease classification using LSTM deep learning approach. CMC-COMPUTERS MATERIALS & CONTINUA, 67(1), 427-443. Storey, Oliver, Bo Wei, Li Zhang, and Franck Romuald Fotso Mtope. "Adaptive bone abnormality detection in medical imagery using deep neural networks." Abreu Dias, D.D., 2019. Musculoskeletal abnormality detection on x-ray using transfer learning.

Round 2

Reviewer 1 Report

The authors have adequately addressed the concerns and the quality of the paper has been improved. Therefore I recommend to accept that paper for publication.

Author Response

Thank you very much for your comment.

Reviewer 2 Report

Authors have addressed all my comments

Author Response

Thank you very much for your comment.

Reviewer 3 Report

The authors did excellent work and resolve almost all my previous queries. These papers will be further improved and interesting for readers more so authors should add few more references in related work such as(AI, Machine and Deep learning Data science)

Uysal, Fatih, Fırat Hardalaç, Ozan Peker, Tolga Tolunay, and Nil Tokgöz. "Classification of Fracture and Normal Shoulder Bone X-Ray Images Using Ensemble and Transfer Learning With Deep Learning Models Based on Convolutional Neural Networks." arXiv preprintarXiv:2102.00515 (2021).Khan, M. A., & Kim, J. (2020). Toward Developing Efficient Conv-AE-Based Intrusion Detection System Using Heterogeneous Dataset. Electronics9(11), 1771.Kegelman, Christopher D., Madhura P. Nijsure, Yasaman Moharrer, Hope B. Pearson, James H. Dawahare, Kelsey M. Jordan, Ling Qin, and Joel D. Boerckel. "YAP and TAZ promote periosteal osteoblast precursor expansion and differentiation for fracture repair." Journal of Bone and Mineral Research 36, no. 1 (2021): 143-157.Sharma, A., Mishra, A., Bansal, A., and Bansal, A., 2021. Bone Fractured Detection Using Machine Learning and Digital Geometry. In Mobile Radio Communications and 5G Networks (pp. 369-376). Springer, Singapore.

Most of the Figures are not the same so all the Fig should be the same and align with text with high resolutions. (Figure 2.)

Author Response

Thank you very much for your comment.

We did not add the study named "Classification of Fracture and Normal Shoulder Bone X-Ray Images Using Ensemble and Transfer Learning With Deep Learning Models Based on Convolutional Neural Networks."

Because this paper is the pre-print of this study you are currently evaluating.

Note: After our study is published, the version in arXiv will be updated by specifying the DOI number.

We added these 3 studies (“Toward Developing Efficient Conv-AE-Based Intrusion Detection System Using Heterogeneous Dataset”, “YAP and TAZ promote periosteal osteoblast precursor expansion and differentiation for fracture repair” , “Bone Fractured Detection Using Machine Learning and Digital Geometry”) in the literature of the related work section.

All figures have been checked and edited to improve resolution and size.